# Active Learning Polynomial Threshold Functions

**Omri Ben-Eliezer**
Department of Mathematics
Massachusetts Institute of Technology
omrib@mit.edu

**Max Hopkins**
Department of Computer Science and Engineering
University of California, San Diego
nmhopkin@eng.ucsd.edu

**Chutong Yang**
Department of Computer Science
Stanford University
yct1998@stanford.edu

**Hantao Yu**
Department of Computer Science
Columbia University
hantao.yu@columbia.edu

## Abstract

We initiate the study of active learning polynomial threshold functions (PTFs). While traditional lower bounds imply that even univariate quadratics cannot be non-trivially actively learned, we show that allowing the learner basic access to the derivatives of the underlying classifier circumvents this issue and leads to a computationally efficient algorithm for active learning degree-$d$ univariate PTFs in $\tilde{O}(d^3 \log(1/\varepsilon\delta))$ queries. We extend this result to the batch active setting, providing a smooth transition between query complexity and rounds of adaptivity, and also provide near-optimal algorithms for active learning PTFs in several average case settings. Finally, we prove that access to derivatives is insufficient for active learning multivariate PTFs, even those of just two variables.

## 1 Introduction

Today's deep neural networks perform incredible feats when provided sufficient training data. Sadly, annotating enough raw data to train your favorite classifier can often be prohibitively expensive, especially in important scenarios like computer-assisted medical diagnoses where labeling requires the advice of human experts. This issue has led to a surge of interest in *active learning*, a paradigm introduced to mitigate extravagant labeling costs. Active learning, originally studied by Angluin in 1988 [1], is in essence formed around two basic hypotheses: raw (unlabeled) data is cheap, and not all data is equally useful. The idea is that by adaptively selecting only the most informative data to label, we can get the same accuracy without the prohibitive cost. As a basic example, consider the class of thresholds in one dimension. Identifying the threshold within some $\varepsilon$ accuracy requires about $1/\varepsilon$ labeled data points, but if we are allowed to *adaptively* select points we can use binary search to recover the same error in only $\log(1/\varepsilon)$ labels, an exponential improvement!

Unfortunately, there's a well-known problem with this approach: it breaks down for most non-trivial classifiers beyond 1D-thresholds [2], providing no asymptotic benefit over standard non-adaptive methods. This has lead researchers in recent years to develop a slew of new strategies overcoming this obstacle. We follow an approach pioneered by Kane, Lovett, Moran, and Zhang (KLMZ) [3]: asking more informative questions. KLMZ suggest that if we are modeling access to a human expert, there's no reason to restrict ourselves to asking only about the labels of raw data; rather, we should be allowed access to other natural application-dependent questions as well. They pay particular attention to learning halfspaces in this model via "comparison queries," which given $x, x' \in \mathbb{R}^d$ ask which point is closer to the bounding hyperplane (think of asking a doctor "which patient is more sick?"). Such queries had already shown promise in practice [4–6], and KLMZ proved they could be used

to efficiently active learn halfspaces in two-dimensions, recovering the exponential improvement seen for 1D-thresholds via binary search. Beyond two dimensions, however, all known techniques either require strong structural assumptions [3, 7], or the introduction of complicated queries [8, 9] requiring infinite precision, a significant limitation in both theory and practice.

The study of active learning halfspaces can be naturally viewed as an attempt to extend the classical active learning of 1D-thresholds to *higher dimensions*. In this work, we take a different approach and instead study the generalization of this problem to *higher degrees*. In particular, we initiate the study of active learning *polynomial threshold functions*, classifiers of the form $\text{sign}(p(x))$ for $x \in \mathbb{R}$ and $p$ some underlying univariate polynomial. When the degree of $p$ is 1, this reduces to the class of 1D-thresholds. Similar to halfspaces, standard arguments show that even degree-two univariate PTFs cannot be actively learned.[1] To this end, we introduce *derivative queries*, a natural class-specific query-type that allows the learner weak access to the derivatives of the underlying PTF $p$.

Derivative queries are well-motivated both in theory and practice. A simple example is the medical setting, where a first-order derivative might correspond to asking "Is patient $X$ recovering, or getting sicker?" Derivatives also play an essential role in our sensory perception of the world. Having two eyes grants us depth perception [10], allowing us to compute low-order derivatives across time-stamps to predict future object positions (e.g. for hunting, collision-avoidance). Multi-viewpoint settings also allow access to low order derivatives by comparing nearby points; one intriguing example is the remarkable sensory echolocation system of bats, which emit ultrasonic waves while moving to learn the structure of their environment [11]. While high order derivatives may be more difficult to compute for a human (or animal) oracle, they still have natural implications in settings such as experimental design where queries are measured mechanically (e.g. automated tests of a self-driving car system might reasonably measure higher order derivatives of positional data). Such techniques have already seen practical success with other query types typically considered too difficult for human annotators (see e.g. the survey of Sverchkov and Craven [12] on automated design in biology).

Our main result can be viewed as theoretical confirmation that this type of question is indeed useful: *derivative queries are necessary and sufficient for active learning univariate PTFs*. We prove that if the learner is allowed access to $\text{sign}(p^{(i)}(x))$ (the $i$-th order derivative of $p$), PTFs are learnable in $O(\log(1/\varepsilon))$ queries, but require $\Omega(1/\varepsilon)$ queries if the learner is missing access even to a single relevant derivative. We generalize this upper bound to the *batch* setting as well, giving a smooth interpolation between query complexity and rounds of communication with data annotators (which have costly overhead in practice).

We also study active learning PTFs beyond the worst-case setting. Specifically, we consider a setup in which the learner is promised that both points in $\mathbb{R}$ and the underlying polynomial are drawn from known distributions. We propose a general algorithm for active learning PTFs in this model based on coupon collecting and binary search, and analyze its query complexity across a few natural settings. Notably, our algorithm in this model avoids the use of derivatives altogether, making it better adapted to scenarios like learning natural imagery where we expect the underlying distributions to be nice, but may not have access to higher order information like derivatives. Finally, we note that all of our upper bounds (in both worst and average-case settings) actually hold for the stronger 'perfect' learning model in which the learner aims to query-efficiently label a fixed 'pool' of data with zero error. Perfect learning is equivalent to active learning in the worst-case model [13, 3], but is likely harder in the average-case and requires new insight over standard techniques in our setting.

We end our work with a preliminary analysis of active learning *multivariate* PTFs, where we prove a strong lower bound showing access to derivative information is actually insufficient to active learn even degree-two PTFs in two variables. We leave upper bounds in this challenging regime (e.g. through distributional assumptions, additional enriched queries) as a direction of future research.

## 1.1 Background

We briefly overview the basic theory of PAC-learning (in both the "passive" and "active" settings) and of the main model we study, *perfect learning*. We cover these topics in much greater detail in the supplementary materials. PAC-learning, originally introduced by Valiant [14] and Vapnik and Chervonenkis [15], provides a framework for studying the learnability of pairs $(X, H)$ where $X$ is

---

[1]By this we mean that adaptivity and the active model provide no asymptotic benefit over the standard "passive" PAC-model.

a set and $H = \{h : X \to \{-1, 1\}\}$ is a family of binary classifiers. A class $(X, H)$ is said to be PAC-learnable in $n = n(\varepsilon, \delta)$ samples if for all $\varepsilon, \delta > 0$, there exists an algorithm $A$ which for all distributions $D$ over $X$ and classifiers $h \in H$, intakes a labeled sample of size $n$ and outputs a good hypothesis with high probability:

$$\Pr_{S \sim D^n}[\mathrm{err}_{D,h}(A(S, h(S))) \leq \varepsilon] \geq 1 - \delta,$$

where $\mathrm{err}_{D,h}(A(S, h(S))) = \mathbb{P}_{x \sim D}[A(S, h(S))(x) \neq h(x)]$. Active learning is a modification of the PAC-paradigm where the learner instead draws *unlabeled* samples, and may choose whether or not they wish to ask for the label of any given point. The goal is to minimize the *query complexity* $q(\varepsilon, \delta)$, which measures the number of queries required to attain the same accuracy guarantees as the standard "passive" PAC-model described above. In the *batch* setting, the learner may send points to the oracle in batches. This incurs the same query cost as in the standard setting (a batch of $m$ points costs $m$ queries), but allows for a finer-grained analysis of adaptivity through the *round complexity* $r(\varepsilon, \delta)$ which measures the total number of batches sent to the oracle.

In this work, we study a challenging variant of active learning called perfect learning (variants of which go by many names in the literature, e.g. RPU-learning [16], perfect selective classification [13], and confident learning [3]).[2] In this model, the learner is asked to label an adversarially selected size-$n$ sample from $X$. The query complexity $q(n)$ (respectively round complexity $r(n)$) is the expected number of queries (respectively rounds) required to infer the labels of all $n$ points in the sample. Perfect learning is well known to be equivalent to active learning up to small factors in query complexity in worst-case settings, and is at least as hard as the latter in the average-case [13, 3]. We discuss these connections in more depth in Section 2.

In this work, we study the learnability of $(\mathbb{R}, H_d)$, the class of degree (at most) $d$ univariate PTFs. In the worst-case setting, we allow the learner access to *derivative queries*: for any $x \in \mathbb{R}$ in the learner's sample, they may query $\mathrm{sign}(f^{(i)}(x))$ for any $i = 0, \ldots, d-1$, where $f^{(i)}$ is the $i$-th derivative of $f$ and $f^{(0)}$ is $f$ itself.

## 1.2 Results

Our main result is that univariate PTFs can be computationally and query-efficiently learned in the perfect model via derivative queries.

**Theorem 1.1** (Perfect Learning PTFs). *The query complexity of perfect learning $(\mathbb{R}, H_d)$ with derivative queries is:*

$$\Omega(d \log n) \leq q(n) \leq O(d^3 \log n).$$

*Furthermore, there is an algorithm achieving this upper bound that runs in time $\tilde{O}(nd)$.*

By standard connections with active learning, this implies PTFs are active learnable in query complexity $\Omega(d \log(1/\varepsilon)) \leq q(\varepsilon, \delta) \leq \tilde{O}\left(d^3 \log(\frac{1}{\varepsilon\delta})\right)$ when the learner has access to derivative queries.

Theorem 1.1 is based on a deterministic algorithm that iteratively learns each derivative given higher order information. This technique necessarily requires a large amount of adaptivity which can be costly in practice. To mitigate this issue, we give a randomized algorithm that extends Theorem 1.1 to the batch setting, providing a smooth trade-off between (expected) query-optimality and adaptivity.

**Theorem 1.2** (Perfect Learning PTFs Batch Setting). *For any $n \in \mathbb{N}$ and $\alpha \in (1/\log(n), 1]$, there exists a randomized algorithm perfectly learning size $n$ subsets of $(\mathbb{R}, H_d)$ in*

$$q(n) \leq O\left(\frac{d^3 n^\alpha}{\alpha}\right)$$

*expected queries, and*

$$r(n) \leq 1 + \frac{2}{\alpha}$$

*expected rounds of adaptivity. Moreover, the algorithm can be implemented in $\tilde{O}(n)$ expected time.*

---

[2]In fact, this model actually precedes active learning, and has long been studied in the computational geometry literature for various concept classes such as halfspaces [17].

When $\alpha = O(1/\log(n))$, this recovers the query complexity of Theorem 1.1 in expectation, but also gives a much broader range of options, e.g. sub-linear query algorithms in $O(1)$ rounds of communication. In fact it is worth noting that even in the former regime the algorithm uses only $O(\log(n))$ total rounds of communication, independent of the underlying PTF's degree. Finally, note that run-time is also near-optimal since $\Omega(n)$ time is required even to read the input.

To complement these upper bounds, we also show that PTFs cannot be actively learned at all if the learner is missing access to any derivative.

**Theorem 1.3** (Perfect Learning PTFs Requires Derivatives). *Any learner using label and derivative queries that is missing access to $f^{(i)}$ for some $1 \le i \le d-1$ must make at least*

$$q(n) \ge \Omega(n)$$

*queries to perfectly learn $(\mathbb{R}, H_d)$.*

This implies the query complexity of active learning PTFs with any missing derivative is $\Omega(1/\varepsilon)$.

In some practical scenarios, our worst-case assumption over the choice of distribution over $\mathbb{R}$ and PTF $h \in H_d$ may be unrealistically adversarial. To this end, we also study a natural average case model for perfect learning, where the sample $S \subset \mathbb{R}$ and PTF $h \in H_d$ are promised to come from known distributions. In Section 4, we show derivates are often unecessary in this regime by providing a generic label-only algorithm and proving query efficiency in basic scenarios. This is better suited than our worst-case analysis to practical scenarios like learning natural 3D-imagery, where we expect objects to come from nice distributions but don't necessarily have higher order information.

We start by considering the basic scenario where both the sample and roots of our PTF are drawn uniformly at random from the interval $[0, 1]$, a distribution we denote by $U_{[0,1]}$.

**Theorem 1.4** (Learning PTFs with Uniformly Random Roots). *The query complexity of perfect learning $(\mathbb{R}, H_d)$ when promised that the sample and roots are chosen from $U_{[0,1]}$ is:*

$$\Omega(d \log n) \le q(n) \le O(d^2 \log d \log n).$$

While studying the uniform distribution is appealing due to its simplicity, similar results can be proved for somewhat more realistic distributions. As an example, we study the case where the (intervals between) roots of our polynomial are drawn from a *Dirichlet distribution* $\mathrm{Dir}(\alpha)$, which has pdf:

$$f(x_1, \ldots, x_{d+1}) \propto \prod_{i=1}^{d+1} x_i^{\alpha-1}$$

where $x_i \ge 0$ and $\sum x_i = 1$. This generalizes drawing a uniformly random point on the $d$-simplex.

**Theorem 1.5** (Learning PTFs with Dirichlet Roots). *The query complexity of perfect learning $(\mathbb{R}, H_d)$ when the subsample $S \sim U_{[0,1]}$ and $h \sim Dir(\alpha)$ is at most*

$$q(n) = O(d^2 \log d \log n) \qquad\qquad \textit{for } \alpha = 1$$
$$q(n) = O(d^2 \log d + d \log n) \qquad\qquad \textit{for } \alpha \ge 2, \textit{ and}$$
$$q(n) = O(d \log n) \qquad\qquad \textit{for } \alpha \ge \Omega(\log^2 n).$$

Moreover, this result is tight for constant $\alpha$ and sufficiently large $n$.

So far we have only discussed *univariate* PTFs. One might reasonably wonder to what extent our results hold for *multivariate* PTFs. In fact, we show that derivative queries are insufficient (in the worst-case setting) for learning PTFs of even two variables.

**Theorem 1.6** (Derivatives Can't Learn Multivariate PTFs). *Let $(\mathbb{R}^2, H_2^2)$ denote the class of degree-two, two-variate PTFs. The query complexity of perfectly learning $(\mathbb{R}^2, H_2^2)$ is*

$$q(n) \ge \Omega(n),$$

*even when the learner may query the sign of the gradient and hessian on any point in its sample.*

In other words, multivariate PTFs cannot be actively learned via access to basic derivative queries in the worst-case. It remains an interesting open problem whether there exist natural query sets that can learn multivariate PTFs, or whether this issue can be avoided in average-case settings; we leave these questions to future work.

## 1.3 Related work

**Active Learning Halfspaces:** While to our knowledge active learning polynomial threshold functions has not been studied in the literature, the closely related problem of learning halfspaces is perhaps one of the best-studied problems in the field, and indeed in learning theory in general. It has long been known that halfspaces cannot be active learned in the standard model [2], but several series of works have gotten around this fact either by restricting the adversary, or empowering the learner. The first of these two methods generally involves forcing the learner to choose a nice marginal distribution over the data, e.g. over the unit sphere [18], unit ball [19], log-concave [20], or more generally $s$-concave distributions [21]. The second approach usually involves allowing the learner to ask some type of additional questions. This encompasses not only KLMZ's [3] notion of enriched queries, but also the original "Membership query" model of Angluin [22] who allowed the learner to query any point in the overall instance space $X$ rather than just on the subsample $S \subset X$. This model is also particularly well-studied for halfspaces where it is called the point-location problem [17, 23–25, 7, 9], and was actually studied originally by Meyer auf der Heide [17] in the perfect learning model even before Angluin's introduction of active learning.

Bounded degree PTFs may be viewed as a special set of halfspaces via the natural embedding to $\{1, x, x^2, \ldots\}$. Given this fact, it is reasonable to ask why our work is not superseded by these prior methods for learning halfspaces. The answer lies in the fact that the query types used in these works are generally very complicated and require infinite precision. For instance, many use arbitrary membership queries (which are known to behave poorly in practice [26]), and even those that sacrifice on query complexity for simpler queries still require arbitrary precision (e.g. the "generalized comparisons" of [8]). Indeed, learning halfspaces even in three dimensions with a simple query set remains an interesting open problem, and our work can be viewed as partial progress in this direction for sets of points that lie on an embedded low-degree univariate polynomial. For instance, one could learn the set $S = \{(x, 3x^5, 5x^7) : x \in [n]\} \subset \mathbb{R}^3$ with respect to any underlying halfspace $\text{sign}(\langle v, \cdot \rangle + b)$ in $O(\log n)$ queries using access to standard labels and the derivatives of the underlying polynomial.

**Active Learning with Enriched Queries:** Our work also fits into a long line of recent studies on learning with enriched queries in theory and in practice. As previously mentioned, Angluin's [22] original membership query model can in a sense be viewed as the seminal work in this direction, and many types of problem-specific enriched queries such as comparisons [27, 4–6, 3, 8, 7, 28, 29, 9, 30], cluster-queries [31–39], mistake queries [40], separation queries [41], and more have been studied since. Along with providing exponential improvements in query complexity in theory, many of these query types have also found use in practice [4, 5, 42, 43, 12]. Indeed even complicated queries such as Angluin's original model that cannot be accurately assessed by humans [26] have found significant use in application to automated experimental design, where the relevant oracle is given by precise scientific measurements rather than a human (see e.g. the seminal work of King et al. "The Automation of Science" [43]). While we view first or second order derivatives as reasonable query types for human experts, higher order derivatives are likely more useful in this latter setting, e.g. in application to dynamical systems where one tracks object movement with physical sensors.

**Average Case Active Learning:** The average-case model we study in this work is the 'perfect' or 'zero-error' variant of the average-case active learning model introduced by Dasgupta [2] (and implicitly in earlier work of Kosaraju, Przytycka, and Borgstrom [44]). These works gave a generic greedy algorithm for active learning finite concept classes $(X, H)$ over arbitrary prior distributions whose query complexity is optimal to within a factor of $O(\log(|H|))$. The exact constants of this approximation were later optimized in the literature on submodular optimization [45], and more recently extended to the batch setting [46]. These works differ substantially from our setting as they focus on giving a generic algorithm for average-case active learning, rather than giving query complexity bounds for any specific class.

Perhaps more similar to our general approach are active learning methods based on Hanneke's disagreement coefficient [47], and Balcan, Hanneke, and Wortman's [48] work on active learning rates over *fixed* instead of worst-case hypotheses. Analysis based on these approaches typically takes advantage of the fact that for a fixed distribution and classifier, the minimum measure of any interval can be considered constant. Our average-case setting can be thought of as a strengthening of this approach in two ways: first we are only promised (weak) concentration bounds on the probability this

measure is small, and second we work in the harder perfect learning model. This latter fact is largely what separates our analysis, as naive attempts at combining prior techniques with concentration lead to 'imperfect' algorithms (ones with a small probability of error). Moving from the low-error to zero-error regime is in general a difficult problem,[3] but is important in high-risk applications like medical diagnoses.[4] Fixing this issue requires analysis of a new 'capped' variant of the coupon collector problem, and proving optimal query bounds requires further involved calculation that would be unnecessary in the low-error active regime.

## 2 Preliminaries

We now cover basic background on learning with enriched queries before sketching the proofs of our main results. Detailed information on all background and full versions of all proofs can be found in the supplementary materials.

**Learning with Enriched Queries:** Recall our learner is allowed to make *derivative queries*, that is given $S \subset \mathbb{R}$, the learner may query $\text{sign}(f^{(i)}(x))$ for any $x \in S, 0 \leq i \leq d - 1$. Such queries have a number of natural interpretations, e.g. the relative distance of objects in image recognition ("is the pedestrian getting closer, or further away?"). Given a PTF $f \in H_d$ and point $x \in S$, it is useful to consider the collection of all derivative queries on $x$ which we call its *sign pattern*.

**Definition 2.1** (Sign Pattern). *The sign pattern of $x \in \mathbb{R}$ with respect to $f \in H_d$ is the vector in* $\{-1, 1\}^{d+1}$:
$$\text{SgnPat}(f, x) = [sign(f(x)), sign(f^{(1)}(x)), \ldots, sign(f^{(d)}(x))].$$

More generally, given a family of binary queries $Q$ (e.g. labels and derivative queries), let $Q_h(T)$ denote the set of all possible query responses to $x \in T$ given $h$ (when $Q$ consists of derivative queries, $Q_h(T)$ is the set of sign patterns in $T$). Since we can rule out any hypotheses $h' \in H$ such that $Q_{h'}(T) \neq Q_h(T)$, we will be interested in the set of *consistent* hypotheses, $H|_{Q_h(T)}$, which satisfy $Q_{h'}(T) = Q_h(T)$. We say that $Q_h(S)$ *infers* the label of a point $x \in X$ when $x$ only has one possible label under the set of consistent hypotheses, that is when for some $z \in \{-1, 1\}$:
$$\forall h' \in H|_{Q_h(S)} : \text{sign}(h')(x) = z.$$

**Inference Dimension:** In their seminal work on the enriched query model, KLMZ [3] introduced *inference dimension*, a combinatorial parameter that exactly characterizes the query complexity of both perfect and active learning under enriched queries.

**Definition 2.2** (Inference Dimension). *The inference dimension of $(X, H)$ with query set $Q$ is the smallest $k$ such that for any subset $S \subset X$ of size $k$, $\forall h \in H$, $\exists x \in S$ s.t. $Q_h(S \setminus \{x\})$ infers $x$. If no such $k$ exists, then we say the inference dimension is $\infty$.*

KLMZ proved that query-efficient learning is possible if and only if inference dimension is finite.

**Theorem 2.3** (Inference Dimension Characterizes Active Learning [3, Theorem 1.5]). *Let $(X, H)$ be a class with inference dimension $k$ with respect to query set $Q$. The expected query complexity of perfectly learning $(X, H)$ is*[5]
$$\Omega(\min(n, k)) \leq q(n) \leq O_k(\log n).$$

*Similarly, the query complexity of active learning $(X, H)$ is at most*
$$q(\varepsilon, \delta) \leq O_k \left( \left( \log \left( \frac{d}{\varepsilon} \right) + \log \left( \frac{1}{\delta} \right) \right) \right),$$

*and if $k = \infty$, active learning gives no asymptotic improvement over standard passive bounds:*
$$q(\varepsilon, \delta) \geq \Omega(1/\varepsilon).$$

---

[3]While the low-error (active) and zero-error (perfect) models are equivalent in the worst-case setting [3], it is not clear whether this is true in average-case settings.

[4]We note that in this setting, the more natural model is Rivest and Sloan's [16] *Reliable and Probably Useful* (RPU) Learning, where the learner can abstain with low probability but may never err. Perfect learning finite samples is essentially equivalent to the RPU model in most settings by standard generalization techniques, including all settings we study.

[5]Formally the upper bound has some dependence on $Q$ as well, which costs an extra factor of $d$ in our setting.

# 3 Worst-Case Active Learning PTFs

## 3.1 Upper Bounds

In this section, we sketch the proofs of our upper bounds Theorem 1.1 and Theorem 1.2. At a high level, both results follow from the fact that a PTF $f$ can be query-efficiently broken into a small number of monotone segments which act like thresholds. This is done in two main steps. First, we observe that it is possible to break $f$ into a small number of segments sharing the same sign pattern.

**Lemma 3.1.** *For any degree-$d$ polynomial $f \in H_d$ and set $S = \{s_1 \leq \ldots \leq s_n\}$, given $\text{sign}(f^{(i)}(x))$ for all $1 \leq i \leq k$ and $x \in S$, it is possible to partition $S$ into $j \leq O(d^2)$ contiguous, disjoint segments*

$$I_1 = [s_1, s_{i_1}], \ I_2 = [s_{i_1+1}, s_{i_2}], \ \ldots, \ I_j = [s_{i_{j-1}+1}, s_n]$$

*such that each interval has a fixed sign pattern, i.e. for every $1 \leq \ell \leq j$ and $s, s' \in I_\ell$:*

$$\text{SgnPat}(s, f^{(1)}) = \text{SgnPat}(s', f^{(1)}).$$

*Moreover, this can be done in $O(n(d + \log n))$ time.*

Second, we observe that $f$ must be monotone on any interval with a fixed pattern.

**Lemma 3.2.** *If $f \in H_d$ and $a < b \in \mathbb{R}$ satisfy $\text{SgnPat}(f^{(1)}, a) = \text{SgnPat}(f^{(1)}, b)$, then $f$ is monotone on $[a, b]$.*

Based on these facts, it is not hard to see that Theorem 1.1 is realized by the following iterative algorithm that learns each derivative in a top-down fashion until reaching $f^{(0)}$ itself.

---

**Algorithm 1:** ITERATIVE-ALGORITHM$(f, S)$

---

**Result:** Label all points in $S$
**Input:** Polynomial $f \in H_d$, Subset $S \subseteq \mathbb{R}$
**Algorithm:**
1 Learn $f^{(d-1)}(x)$ by binary search
2 $i \leftarrow d - 2$
3 **while** $i \geq 0$ **do**
4 $\quad$ Apply Lemma 3.1 to $f^{(i)}$, partitioning $S$ into $j \leq O((d-i)^2)$ monotone segments $\{I_\ell\}$
5 $\quad$ Learn $f^{(i)}$ by running binary search separately on each $I_\ell$
6 $\quad$ $i \leftarrow i - 1$
7 **end**

---

The iterative approach gives a simple, deterministic technique for learning PTFs with derivative queries, but comes at the cost of a high level of adaptivity. We now give a simple randomized algorithm that smoothly interpolates between the query-efficient and low-adaptivity regimes.

Algorithm 2 is a batch variant of KLMZ's original algorithm underlying Theorem 2.3. The following batch variant of their upper bound follows from similar analysis (setting $m$ above to $2kn^\alpha$).

**Theorem 3.3** (Inference Dimension $\rightarrow$ Batch Active Learning). *Let $(X, H)$ be a class with inference dimension $k$ with respect to query set $Q$. Then for any $n \in \mathbb{N}$ and $\alpha \in (1/\log(n), 1]$ and size $n$-subset $S \subset X$, BATCH-KLMZ labels all of $S$ in*

$$q(n) \leq \frac{2Q_{tot}(2kn^\alpha)}{\alpha}$$

*expected queries, and only*

$$r(n) \leq 1 + \frac{2}{\alpha}$$

*expected rounds of adaptivity, where $Q_{tot}(m)$ is the number of queries available on a set of $m$ points.*

To prove Theorem 1.2, it therefore suffices to show that PTFs have bounded inference dimension with respect to derivative queries. In fact, this is essentially immediate from Lemma 3.1 and Lemma 3.2. Any $f \in H_d$ can be broken up into $O(d^2)$ monotonic regions based on sign pattern. Given any three points in such a region the middle can always be inferred, so by the pigeonhole principle we have:

**Lemma 3.4.** *The inference dimension of $(\mathbb{R}, H_d)$ with derivative queries is $O(d^2)$.*

In the supplementary materials, we additionally prove an $\Omega(d)$ lower bound on inference dimension.

**Algorithm 2:** BATCH-KLMZ$(S, m)$

---

**Result:** Labels all points in $S$
**Input:** Class $(X, H)$, Subset $S \subseteq X$, Query set $Q$, Query Oracle $O_Q$
**Parameters:** Inference dimension $k$, Batch size $m$, Iteration cutoff $t = \frac{\log(n)}{\log(\frac{m}{2k})}$
**Algorithm:**

8   $S_0 \leftarrow S$
9   **for** *i in range t* **do**
10     $T \leftarrow \{\}$
11     **while** $Q_h(T)$ *infers less than a* $\frac{m-2k}{m}$ *fraction of* $S_i$ **do**
12       Sample $T \sim S_i^m$
13       Query $T$: $Q_h(T) \leftarrow O_Q(T)$
14     **end**
15     $S_{i+1} \leftarrow \{x \in S_i : Q_h(T) \text{ does not infer } x\}$
16     **if** $|S_{i+1}| \leq m$ **then**
17       Query $O_Q(S_{i+1})$
18       **Return**
19     **end**
20 **end**

---

## 3.2 Lower Bounds

We briefly discuss the proof techniques behind our lower bounds in Theorem 1.1 and Theorem 1.3. The former follows from a standard information theoretic argument, noting that degree-$d$ PTFs result in $n^{\Omega(d)}$ possible labelings of any $n$ point subset. For the latter, let $Q_{\hat{i}}$ denote the family of derivative queries without the $i$th derivative. Theorem 2.3 shows it is enough to prove the inference dimension of $(\mathbb{R}, H_d)$ with respect to $Q_{\hat{i}}$ is infinite. By a simple induction, it is sufficient to consider $i = d - 1$. This is done by exhibiting for all $n \in \mathbb{N}$ a family of PTFs $\{h, p_1, \ldots, p_n\}$ and points $S = \{s_1, \ldots, s_n\}$ such that $h$ and $p_j$ are indistinguishable on $S \setminus \{s_j\}$ with respect to $\hat{Q}_{d-1}$, but $h(s_j) \neq p_j(s_j)$. We achieve such a construction by building a set of polynomials where each $p_i^{(d-1)}$ is sufficiently negative around $s_i$ to force the function to flip sign, but the points are sufficiently spread out that this cannot be detected on any other $s_j$ for $j \neq i$.

## 4 Average-Case Active Learning PTFs

We now sketch the proof of our average-case results. We briefly recall the model: given distributions $D_X$ over $\mathbb{R}$ and $D_H$ over $H_d$, we are interested in analyzing the expected number of queries needed to infer all labels of a sample $S \sim D_X$ with respect to $h \sim D_H$. We now present a simple generic algorithm for this problem we call "Sample and Search." Given a sample $S \subset \mathbb{R}$:

1. Query the label (sign) of points from $S$ uniformly at random until either:
   (a) We have queried all $n$ points in $S$.
   (b) We see $d$ sign flips in the queried points, i.e. we have queried $x_1, \ldots, x_k$ and there exists indices $i_1 < \ldots < i_{d+1}$ such that
   
   $$\text{sign}(f(x_{i_j})) \neq \text{sign}(f(x_{i_{j+1}}))$$
   
   for all $j = 1, \ldots, d$.
2. If (b) occurred in the previous step, perform binary search on the points in $S$ between each pair $(x_{i_j}, x_{i_{j+1}})$ to find the sign threshold (and thereby labels) in that interval.

It is not hard to see Sample and Search infers all labels of points in $S$ by construction. The main challenge lies in analyzing its query complexity, and in particular step 1 which can be thought of as a variant of the classical coupon collector (CC) problem. In our setting, the "coupons" are made up by the intervals between adjacent roots, and their probability is given by the mass of the marginal distribution on that interval. With this in mind, let $Y$ be the random variable measuring the number of samples required to hit each interval (coupon) at least once, and let $Z = \min(Y, n)$.

**Proposition 4.1.** *The expected query complexity of the Sample and Search Algorithm is at most:*

$$q(n) \leq \mathbb{E}_{D_X, D_H}[Z] + d \log n.$$

It is worth noting that $\mathbb{E}_{D_X, D_H}[Z]$ and $\mathbb{E}_{D_X, D_H}[Y]$ can differ drastically. As a basic example, consider the case where $d = 1$ and we draw our $n$ points and one root uniformly at random from $[0, 1]$. It is a simple exercise to show that $\mathbb{E}_{D_X, D_H}[Y] = \infty$, whereas $\mathbb{E}_{D_X, D_H}[Z] = O(\log n)$. With this in mind, we finish the section by sketching the proof of our average-case results. We focus on the setting of the uniform distribution (Theorem 1.4). The Dirichlet case (Theorem 1.5) follows similar overall ideas, but requires more involved calculation and machinery to deal with dependence of the roots. In both cases, however, the first step is to observe the following standard bound on $Y$ conditional on the roots being well separated.

**Lemma 4.2.** *For any $x \in \mathbb{R}_+$, let $E_x$ denote the event that $f \sim D_H$ has measure at least $\frac{1}{x}$ over $D_X$ between any two adjacent roots, the leftmost root and $0$, and the rightmost root and $1$. Then:*

$$\mathbb{E}_{D_X, D_H}[Y | E_x] \leq O(x \log d).$$

To analyze the capped variable $Z = \min\{Y, n\}$, we expand the expectation and cap the integral:

$$\mathbb{E}_{D_X, D_H}[Z] = \int_0^\infty 1 - \mathbb{P}_{D_H}[\mathbb{E}_{D_X}[Z] \leq x] dx \leq (d+1) + \int_{d+1}^n 1 - \mathbb{P}_{D_H}[\mathbb{E}_{D_X}[Y] \leq x] dx. \quad (1)$$

By Lemma 4.2, the righthand probability is lower bounded by the probability the minimum interval measure (denoted M) is $\Omega(\frac{\log d}{x})$, which can be computed directly in the uniform setting:

$$\mathbb{P}_{D_H}[\mathbb{E}_{D_X}[Y] \leq x] \geq \mathbb{P}\left[M \geq \frac{c \log d}{x}\right] = \left(1 - \frac{c(d-1) \log d}{x}\right)^d. \quad (2)$$

Plugging this back into Equation (1) and computing the integral gives $\mathbb{E}[Z] \leq O(d^2 \log(n))$. To prove the corresponding lower bound for this problem (appearing in Theorem 1.4), we appeal to classic information theoretic techniques. In particular, the expected number of binary queries to reveal the labels of $S$ cannot be less than the entropy of the resulting distribution over labelings. In the uniform case, one can directly show the entropy is $\Omega(d \log(n))$, so Sample and Search is off from the information theoretic optimum by at most a factor of $d$.

We briefly remark on the challenges moving to the Dirichlet distribution. Due to the dependence of roots in this setting, bounding the minimum measure $M$ in Equation (2) becomes substantially more difficult. For small $\alpha$, we union bound over each variable and directly analyze the marginal distributions. For large $\alpha$, we appeal to strong Bernstein-type concentration bounds on Beta distributions of Skorski [49]. The lower bound follows from a similar appeal to information theoretic techniques. The trick in this case is to observe the induced label distribution is exactly the well-studied "Dirichlet-Multinomial" distribution whose asymptotic entropy is known (see e.g. [50, Theorem 2]).

## 5 Beyond Univariate PTFs

Finally, we sketch the proof of Theorem 1.6, our lower bound on learning two-variate quadratics with derivative queries. Recall in this setting the learner is allowed to make *gradient queries* of the form $\text{sign}\left(\frac{\partial f}{\partial x}(x_1, y_1), \frac{\partial f}{\partial y}(x_1, y_1)\right)$, and *Hessian queries* of the form $\text{sign}\left(\frac{\partial^2 f}{\partial x \partial x}(x_1, y_1), \frac{\partial^2 f}{\partial x \partial y}(x_1, y_1), \frac{\partial^2 f}{\partial y \partial x}(x_1, y_1), \frac{\partial^2 f}{\partial y \partial y}(x_1, y_1)\right)$ for any $(x_1, y_1) \in \mathbb{R}^2$ in the learner's sample (along with standard label queries).

As in our worst-case lower bound for missing derivatives, we prove the inference dimension of this class is infinite. Our construction consists of $n$ points distributed evenly on the quarter circle of radius 1 in the first quadrant. Consider the functions $-x^2 - y^2 \pm \epsilon xy$ (with $\epsilon$ small enough), which have negative values for all label queries, gradient queries, and diagonal entries of the Hessian, and either all positive or all negative values for the off-diagonal. The goal is then to find functions $h_i$ which flip the label on the $i$th point, but are indistinguishable from one of $-x^2 - y^2 \pm \epsilon xy$ elsewhere (at least half the points are therefore indistinguishable from one of these choices, which gives an inference dimension lower bound of $n/2$ for any $n \in \mathbb{N}$).

The idea is to consider rotations of the function $f = xy - c_1 y^2$, which is only positive for a small sector dependent on $c_1$. $h_i$ is chosen to be the rotation where this sector contains only the $i$th point. To ensure the derivatives and diagonal Hessian values remain negative, we subtract $c_2(x^2 + y^2 - 1)$ which does not change the value on the unit circle. Since the Hessian is symmetric for these functions, they also agree with one of $-x^2 - y^2 \pm \epsilon xy$ on the off-diagonal giving the desired result.

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
