# Active Learning Polynomial Threshold Functions

Omri Ben-Eliezer*    Max Hopkins†    Chutong Yang‡    Hantao Yu§

October 1, 2022

## Abstract

We initiate the study of active learning polynomial threshold functions (PTFs). While traditional lower bounds imply that even univariate quadratics cannot be non-trivially actively learned, we show that allowing the learner basic access to the derivatives of the underlying classifier circumvents this issue and leads to a computationally efficient algorithm for active learning degree-$d$ univariate PTFs in $\tilde{O}(d^3 \log(1/\varepsilon\delta))$ queries. We extend this result to the batch active setting, providing a smooth transition between query complexity and rounds of adaptivity, and also provide near-optimal algorithms for active learning PTFs in several average case settings. Finally, we prove that access to derivatives is insufficient for active learning multivariate PTFs, even those of just two variables.

## 1 Introduction

Today's deep neural networks perform incredible feats when provided sufficient training data. Sadly, annotating enough raw data to train your favorite classifier can often be prohibitively expensive, especially in important scenarios like computer-assisted medical diagnoses where labeling requires the advice of human experts. This issue has led to a surge of interest in *active learning*, a paradigm introduced to mitigate extravagant labeling costs. Active learning, originally studied by Angluin in 1988 [1], is in essence formed around two basic hypotheses: raw (unlabeled) data is cheap, and not all data is equally useful. The idea is that by adaptively selecting only the most informative data to label, we can get the same accuracy without the prohibitive cost. As a basic example, consider the class of thresholds in one dimension. Identifying the threshold within some $\varepsilon$ accuracy requires about $1/\varepsilon$ labeled data points, but if we are allowed to *adaptively* select points we can use binary search to recover the same error in only $\log(1/\varepsilon)$ labels, an exponential improvement!

Unfortunately, there's a well-known problem with this approach: active learning actually breaks down for most non-trivial classifiers beyond 1D-thresholds [2], providing no asymptotic benefit over standard non-adaptive methods. This has lead researchers in recent years to develop a slew of new strategies overcoming this obstacle. We follow an approach pioneered by Kane, Lovett, Moran, and Zhang (KLMZ) [3]: asking more informative questions. KLMZ suggest that if we are modeling access to a human expert, there's no reason to restrict ourselves to asking only about the labels of raw data; rather, we should be allowed access to other natural application-dependent questions as well. They pay particular attention to learning halfspaces in this model via "comparison queries," which given $x, x' \in \mathbb{R}^d$ ask which point is closer to the bounding hyperplane (think of asking a doctor "which patient is more sick?"). Such queries had already shown promise in practice [4, 5, 6], and KLMZ proved they could be used to efficiently active learn halfspaces in two-dimensions, recovering the exponential improvement seen for 1D-thresholds via binary search. Beyond two dimensions, however, all known techniques either require strong structural assumptions [3, 7], or the introduction of complicated queries [8, 9] requiring infinite precision, a significant limitation in both theory and practice.

---

*Department of Mathematics, MIT, MA 02139. Email: `omrib@mit.edu`.

†Department of Computer Science and Engineering, UCSD, CA 92093. Email: `nmhopkin@eng.ucsd.edu`. Supported by NSF Award DGE-1650112.

‡Department of Computer Science, Stanford University, CA 94305. Email: `yct1998@stanford.edu`

§Department of Computer Science, Columbia University, NY 10027. Email: `hy2751@columbia.edu`.

The study of active learning halfspaces can be naturally viewed as an attempt to extend the classical active learning of 1D-thresholds to *higher dimensions*. In this work, we take a somewhat different approach and instead study the generalization of this problem to *higher degrees*. In particular, we initiate the study of active learning *polynomial threshold functions*, classifiers of the form $\text{sign}(p(x))$ for $x \in \mathbb{R}$ and $p$ some underlying univariate polynomial. When the degree of $p$ is 1, this reduces to the class of 1D-thresholds. Similar to halfspaces, standard arguments show that even degree-two univariate PTFs cannot be actively learned.[1] To this end, we introduce *derivative queries*, a natural class-specific query-type that allows the learner weak access to the derivatives of the underlying PTF $p$.

Derivative queries are well-motivated both in theory and practice. A simple example is the medical setting, where a first-order derivative might correspond to asking "Is patient $X$ recovering, or getting sicker?" Derivatives also play an essential role in our sensory perception of the world. Having two eyes grants us depth perception [10], allowing us to compute low-order derivatives across time-stamps to predict future object positions (e.g. for hunting, collision-avoidance). Multi-viewpoint settings also allow access to low order derivatives by comparing nearby points; one intriguing example is the remarkable sensory echolocation system of bats, which emit ultrasonic waves while moving to learn the structure of their environment [11]. While high order derivatives may be more difficult to compute for a human (or animal) oracle, they still have natural implications in settings such as experimental design where queries are measured mechanically (e.g. automated tests of a self-driving car system might reasonably measure higher order derivatives of positional data). Such techniques have already seen practical success with other query types typically considered too difficult for human annotators (see e.g. the survey of Sverchkov and Craven [12] on automated design in biology).

Our main result can be viewed as a theoretical confirmation that this type of question is indeed useful: *derivative queries are necessary and sufficient for active learning univariate PTFs*. In slightly more detail, we prove that if a learner is allowed access to $\text{sign}(p^{(i)}(x))$, PTFs are learnable in $O(\log(1/\varepsilon))$ queries. On the other hand, if the learner is missing access to even a single relevant derivative, active learning becomes impossible and the complexity returns to the standard $\Omega(1/\varepsilon)$ lower bound. We generalize this upper bound to the popular *batch* active setting as well, giving a smooth interpolation between query complexity and total rounds of communication with data annotators (which can have costly overhead in practice).

We also study active learning PTFs beyond the worst-case setting. Specifically, we consider a setup in which the learner is promised that both points in $\mathbb{R}$ and the underlying polynomial are drawn from known underlying distributions. We propose a general algorithm for active learning PTFs in this model based on coupon collecting and binary search, and analyze its query complexity across a few natural settings. Notably, our algorithm in this model avoids the use of derivatives altogether, making it better adapted to scenarios like learning 3D-imagery where we expect the underlying distributions to be natural or structured, but may not have access to higher order information like derivatives. We note that all of our upper bounds (in both worst and average-case settings) actually hold for the stronger 'perfect' learning model in which the learner aims to query-efficiently label a fixed 'pool' of data with zero error. Perfect learning is equivalent to active learning in the worst-case setting [13, 3], but is likely harder in the average-case and requires new insight over standard techniques.

Finally, we end our work with a preliminary analysis of active learning *multivariate* PTFs, where we prove a strong lower bound showing that access to derivative information is actually insufficient to active learn even degree-two PTFs in two variables. We leave upper bounds in this more challenging regime (e.g. through distributional assumptions or additional enriched queries such as comparisons) as an interesting direction of future research.

## 1.1  Background

Before delving into our results, we briefly overview the basic theory of PAC-learning (in both the "passive" and "active" settings) and of the main model we study, *perfect learning*. We cover these topics in much greater detail in Section 2. PAC-learning, originally introduced by Valiant [14] and Vapnik and Chervonenkis [15], provides a framework for studying the learnability of pairs $(X, H)$ where $X$ is a set and $H = \{h : X \to$

---

[1]By this we mean that adaptivity and the active model provide no asymptotic benefit over the standard "passive" PAC-model.

$\{-1, 1\}\}$ is a family of binary classifiers. A class $(X, H)$ is said to be PAC-learnable in $n = n(\varepsilon, \delta)$ samples if for all $\varepsilon, \delta > 0$, there exists an algorithm $A$ which for all distributions $D$ over $X$ and classifiers $h \in H$, intakes a labeled sample of size $n$ and outputs a good hypothesis with high probability:

$$\Pr_{S \sim D^n}[\text{err}_{D,h}(A(S, h(S))) \leq \varepsilon] \geq 1 - \delta,$$

where $\text{err}_{D,h}(A(S, h(S))) = \mathbb{P}_{x \sim D}[A(S, h(S))(x) \neq h(x)]$. Active learning is a modification of the PAC-paradigm where the learner instead draws *unlabeled* samples, and may choose whether or not they wish to ask for the label of any given point. The goal is to minimize the *query complexity* $q(\varepsilon, \delta)$, which measures the number of queries required to attain the same accuracy guarantees as the standard "passive" PAC-model described above. In the *batch* setting, the learner may send points to the oracle in batches. This incurs the same query cost as in the standard setting (a batch of $m$ points costs $m$ queries), but allows for a finer-grained analysis of adaptivity through the *round complexity* $r(\varepsilon, \delta)$ which measures the total number of batches sent to the oracle.

In this work, we study a challenging variant of active learning called perfect learning (variants of which go by many names in the literature, e.g. RPU-learning [16], perfect selective classification [13], and confident learning [3]).[2] In this model, the learner is asked to label an adversarially selected size-$n$ sample from $X$. The query complexity $q(n)$ (respectively round complexity $r(n)$) is the expected number of queries (respectively rounds) required to infer the labels of all $n$ points in the sample. Perfect learning is well known to be equivalent to active learning up to small factors in query complexity in worst-case settings, and is at least as hard as the latter in the average-case. We discuss these connections in more depth in Section 2.

In this work, we study the learnability of $(\mathbb{R}, H_d)$, the class of degree (at most) $d$ univariate PTFs. In the standard worst-case settings described above, we will allow the learner access to *derivative queries*, that is, for any $x \in \mathbb{R}$ in the learner's sample, they may query $\text{sign}(f^{(i)}(x))$ for any $i = 0, \dots, d-1$, where $f^{(i)}$ is the $i$-th derivative of $f$.

## 1.2 Results

Our main result is that univariate PTFs can be computationally and query-efficiently learned in the perfect model via derivative queries.

**Theorem 1.1** (Perfect Learning PTFs (Theorem 3.2)). *The query complexity of perfect learning $(\mathbb{R}, H_d)$ with derivative queries is:*

$$\Omega(d \log n) \leq q(n) \leq O(d^3 \log n).$$

*Furthermore, there is an algorithm achieving this upper bound that runs in time $\tilde{O}(nd)$.*

Note that by standard connections with active learning, this implies that PTFs are actively learnable with query complexity $\Omega(d \log(1/\varepsilon)) \leq q(\varepsilon, \delta) \leq \tilde{O}\left(d^3 \log(\frac{1}{\varepsilon \delta})\right)$ when the learner has access to derivative queries.

Theorem 1.1 is based on a deterministic algorithm that iteratively learns each derivative given higher order information. This technique necessarily requires a large amount of adaptivity which can be costly in practice. To mitigate this issue, we also give a simple randomized algorithm that extends Theorem 1.1 to the batch setting and provides a smooth trade-off between (expected) query-optimality and adaptivity.

**Theorem 1.2** (Perfect Learning PTFs Batch Setting (Theorem 3.5)). *For any $n \in \mathbb{N}$ and $\alpha \in (1/\log(n), 1]$, there exists a randomized algorithm perfectly learning size $n$ subsets of $(\mathbb{R}, H_d)$ in*

$$q(n) \leq O\left(\frac{d^3 n^\alpha}{\alpha}\right)$$

*expected queries, and*

$$r(n) \leq 1 + \frac{2}{\alpha}$$

---

[2]In fact, this model actually precedes active learning, and has long been studied in the computational geometry literature for various concept classes such as halfspaces [17].

*expected rounds of adaptivity. Moreover, the algorithm can be implemented in $\tilde{O}(n)$ expected time.*

When $\alpha = O(1/\log(n))$, this recovers the query complexity of Theorem 1.1 in expectation, but also gives a much broader range of options, e.g. sub-linear query algorithms in $O(1)$ rounds of communication. In fact it is worth noting that even in the former regime the algorithm uses only $O(\log(n))$ total rounds of communication, independent of the underlying PTF's degree. Finally, note that the run-time is also near-optimal since there is a trivial lower bound of $\Omega(n)$ required even to read the input.

To complement these upper bounds, we also show that PTFs cannot be actively learned at all if the learner is missing access to any derivative.

**Theorem 1.3** (Perfect Learning PTFs Requires Derivatives (Theorem 3.9)). *Any learner using label and derivative queries that is missing access to $f^{(i)}$ for some $1 \leq i \leq d-1$ must make at least*

$$q(n) \geq \Omega(n)$$

*queries to perfectly learn $(\mathbb{R}, H_d)$.*

Similarly, this implies the query complexity of active learning PTFs with any missing derivative is $\Omega(1/\varepsilon)$.

In some practical scenarios, our worst-case assumption over the choice of distribution over $\mathbb{R}$ and PTF $h \in H_d$ may be unrealistically adversarial. To this end, we also study a natural average case model for perfect learning, where the sample $S \subset \mathbb{R}$ and PTF $h \in H_d$ are promised to come from known distributions. In Section 4, we discuss a fairly general algorithm for this regime based on combining a randomized variant of coupon collecting with binary search. As applications, we analyze the query complexity of learning $(\mathbb{R}, H_d)$ in several basic distributional settings, and show that derivative queries are actually unnecessary for optimal active learning in the distributional setting.

We start by considering the basic scenario where both the sample and roots of our PTF are drawn uniformly at random from the interval $[0, 1]$, a distribution we denote by $U_{[0,1]}$.

**Theorem 1.4** (Learning PTFs with Uniformly Random Roots (Theorem 4.3)). *The query complexity of perfect learning $(\mathbb{R}, H_d)$ when promised that the sample and roots are chosen from $U_{[0,1]}$ is:*

$$\Omega(d \log n) \leq q(n) \leq \tilde{O}(d^2 \log n).$$

While studying the uniform distribution is appealing due to its simplicity, similar results can be proved for other, perhaps more practically realistic distributions. As an example, we study the case where the (intervals between) roots of our polynomial are drawn from a *Dirichlet distribution* $\mathrm{Dir}(\alpha)$, which has pdf:

$$f(x_1, \ldots, x_{d+1}) \propto \prod_{i=1}^{d+1} x_i^{\alpha-1}$$

where $x_i \geq 0$ and $\sum x_i = 1$. This generalizes drawing a uniformly random point on the $d$-simplex.

**Theorem 1.5** (Learning PTFs with Dirichlet Roots (Theorem 4.7)). *The query complexity of perfect learning $(\mathbb{R}, H_d)$ when the subsample $S \sim U_{[0,1]}$ and $h \sim Dir(\alpha)$ is at most*

$$q(n) \leq \tilde{O}(d^2 \log n)$$

*when $\alpha = 1$,*

$$q(n) \leq \tilde{O}(d^2 + d \log n)$$

*when $\alpha \geq 2$, and*

$$q(n) \leq O(d \log n)$$

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

## 1.4 Roadmap

The remainder of this paper proceeds as follows: in Section 2 we cover background and preliminary definitions, in Section 3 we characterize the active learnability of PTFs with derivative queries in the worst-case standard and batch models, in Section 4 we discuss active learning PTFs in average-case settings, and in Section 5 we prove that derivative queries and Hessian queries are insufficient for active learning multivariate PTFs.

# 2 Preliminaries

Before moving on to our main results, we cover some preliminary background on PAC-learning, introduce the perfect learning model and its relation to active learning, and discuss enriched queries along with KLMZ's related notion of inference dimension.

## 2.1 PAC-Learning

A *hypothesis class* consists of a pair $(X, H)$ where $X$ is a set called the *instance space* and $H = \{h : X \to \{-1, 1\}\}$ is a family of binary classifiers. We call each $h \in H$ a *hypothesis*. In this paper, we study hypothesis classes of the form:
$$H = \{\text{sign}(f) : f \in \mathcal{F}\},$$
where $\mathcal{F} = \{f : X \to \mathbb{R}\}$ is a family of real-valued functions over $X$, and $\text{sign}(f)$ is defined as $\text{sign}(f)(x) = \text{sign}(f(x))$ for all $x \in X$.[5] When clear from context, we will often refer to classifiers in $H$ by their underlying function in $\mathcal{F}$.

---

[3]While the low-error (active) and zero-error (perfect) models are equivalent in the worst-case setting [3], it is not clear whether this is true in average-case settings.

[4]We note that in this setting, the more natural model is Rivest and Sloan's [16] *Reliable and Probably Useful* (RPU) Learning, where the learner can abstain with low probability but may never err. Perfect learning finite samples is essentially equivalent to the RPU model in most settings by standard generalization techniques, including all settings we study.

[5]We adopt the standard convention $\text{sign}(0) = 1$ in this work, though $\text{sign}(0) = -1$ works equally well in all our arguments.

An *example* is a pair $(x, y) \in X \times \{1, -1\}$. A *labeled sample* $\bar{S}$ is a finite sequence of examples, and we can remove all labels of $\bar{S}$ to get the corresponding *unlabeled sample* $S$. Given a distribution $D$ on $X \times \{1, -1\}$, the *expected loss* of a hypothesis $h$ is

$$L_D(h) = \mathop{\mathbb{P}}_{(x,y) \sim D}[h(x) \neq y].$$

A distribution $D$ on $X \times \{1, -1\}$ is *realizable* by $H$ if there exists $h \in H$ such that $L_D(h) = 0$.

A *learning algorithm* takes a labeled sample as input, and outputs a hypothesis. Following the model of Valiant [14] and Vapnik-Chervonenkis [49], we say a class $(X, H)$ is *PAC-learnable* in sample complexity $n = n(\varepsilon, \delta)$ if for all $\varepsilon, \delta > 0$ there exists a learning algorithm $A$ which outputs a good hypothesis with high probability over samples $\bar{S} \sim D^n$ from any realizable distribution $D$:

$$\mathop{\mathbb{P}}_{\bar{S} \sim D^n}[L_D(h) > \varepsilon] \leq \delta.$$

PAC-learning is well-known to be characterized by a combinatorial parameter called VC-dimension. Namely, the sample complexity of learning a class of VC-dimension $d$ is about $n(\varepsilon, \delta) = \tilde{\Theta}(\frac{d + \log(1/\delta)}{\varepsilon})$ [50].

## 2.2 Active Learning

Unfortunately, in practice it is often the case that obtaining enough labeled data to PAC-learn is prohibitively expensive. This motivates the study of *active learning*, a model in which the algorithm is provided an *unlabeled* sample $S$ along with access to a labeling oracle it can query for the label of any $x \in S$. In this setting, our goal is generally to minimize the number of queries made to the oracle while maintaining PAC-learning guarantees. We say a class $(X, H)$ is actively PAC-learnable in sample complexity $n = n(\varepsilon, \delta)$ and query complexity $q = q(\varepsilon, \delta)$ if for all $\varepsilon, \delta > 0$ there exists a learning algorithm $A$ which outputs a good hypothesis with high probability over samples $\bar{S} \sim D^n$ from any realizable distribution $D$:

$$\mathop{\mathbb{P}}_{S \sim D^n}[L_D(h) > \varepsilon] \leq \delta,$$

and makes at most $q(\varepsilon, \delta)$ queries. In this paper we will focus mostly on the query complexity $q(\varepsilon, \delta)$. Note that the goal in active learning is generally to have $q(\varepsilon, \delta)$ be around $\log(n(\varepsilon, \delta)) \approx \log(1/\varepsilon)$, and that this is easy to achieve for very basic classes like 1D-thresholds (e.g. by binary search). It is not hard to see that $\Omega(\log(1/\varepsilon))$ queries is information theoretically optimal for most non-trivial hypothesis classes, as the bound follows from identifying a polynomially-sized $\Omega(\varepsilon)$-packing [51] (which is generally easy to do for non-trivial classes).

Recent years have also seen an increased interest in *batch* active learning, a model which takes into account the high overhead of sending and receiving data from the labeling oracle. In this model, the learner may send points to the oracle in *batches*. Query complexity is measured the same as in the standard model (sending a batch of $m$ points still incurs $m$ cost in query complexity), but algorithms are additionally parametrized by their *round complexity* $r(\varepsilon, \delta)$, which denotes the total number of times the learner sent batches to the oracle. In practice, it is often more efficient to sacrifice some amount of query efficiency in order to reduce the round complexity and its associated overhead cost. Theoretically, the round complexity acts as a measure of total adaptivity, interpolating between the passive PAC regime (where $r(\varepsilon, \delta) = 1$), and the active regime (where $r(\varepsilon, \delta) = q(\varepsilon, \delta)$).

## 2.3 Learning with Enriched Queries

Unfortunately, beyond basic classes such as thresholds, even the full adaptivity of the standard active model generally fails to provide any asymptotic improvement over the passive learning (even for basic extensions such as halfspaces in two dimensions [2]). To circumvent this issue, instead of querying only labels, we consider learners which can ask other natural questions about the data as well. In this work, we mainly focus on the hypothesis class $(\mathbb{R}, H_d)$, where $H_d$ is the set of univariate degree (at most) $d$ polynomials over $\mathbb{R}$. Since this class is not actively learnable in the traditional model, we will allow our learners access to

the derivatives of $f \in H_d$ in the following sense: given an unlabeled sample $S \subset \mathbb{R}$, the learner may query $\text{sign}(f^{(i)}(x))$ for any $x \in S$, $0 \leq i \leq d-1$, which we call *derivative queries*. In the introduction we discussed a practical interpretation of derivative queries in the medical domain. Another natural interpretation might be in image recognition, where such a query could correspond to the relative distance of an object from the observer ("is the pedestrian getting closer, or further away?"). While higher order derivatives may be difficult for humans to measure in such applications, they can certainly be recorded by physical sensors, e.g. in the dash-cam of a self-driving car.

Given such an $f \in H_d$ and $x \in S$, it will be useful to consider the collection of all derivative queries on $x$, an object we call $x$'s *sign pattern*.

**Definition 2.1** (Sign Pattern). *The sign pattern of $x \in \mathbb{R}$ with respect to $f \in H_d$ is the vector in $\{-1, 1\}^{d+1}$:*

$$\text{SgnPat}(f, x) = [sign(f(x)), sign(f^{(1)}(x)), \ldots, sign(f^{(d)}(x))].$$

More generally, given a family of binary queries $Q$ (e.g. labels and derivative queries), let $Q_h(T)$ denote the set of all possible query responses to $x \in T$ given $h$ (so when $Q$ consists of labels and derivative queries, $Q_h(T)$ is just the set of sign patterns for each $x \in T$ under $h$). Notice that given such a query response, we can rule out any hypotheses $h' \in H$ such that $Q_{h'}(T) \neq Q_h(T)$. As a result, we will be interested in the set of *consistent* hypotheses, $H|_{Q_h(T)}$, which satisfy $Q_{h'}(T) = Q_h(T)$. Finally, since our overall goal is to learn the labels of elements in $X$, we will be interested in when a query response $Q_h(S)$ *infers* the label of a point $x \in X$. Formally, this occurs when $x$ only has one possible label under the set of consistent hypotheses:

$$\forall h' \in H|_{Q_h(S)} : Q_{h'}(x) = z$$

where $z \in \{-1, 1\}$.

## 2.4 Perfect Learning

In this work, we will study a slightly stronger model of active learning called *perfect* or *confident* learning. In this setting, the learner is given an arbitrary finite sample $S \subset \mathbb{R}$, and must infer the labels under an adversarially chosen classifier. Variants of this model have been studied in the computational geometry [17, 23, 24, 25, 9], statistical learning theory [16, 13, 3, 7, 28], and clustering literatures [31, 39] under various names. Formally, we say a class $(X, H)$ is *perfectly learnable* with respect to a query set $Q$ in $q(n)$ expected queries if there exists an algorithm $A$ such that for every $n \in \mathbb{N}$, every sample $S \subset X$ of size $n$, and every hypothesis $h \in H$, $A$ correctly labels all of $S$ with respect to $h$ in at most $q(n)$ queries in expectation over the internal randomness of the algorithm. In the batch model, query and round complexity are defined analogously.

Since worst-case guarantees are often too strict in practice, we will also study an average-case variant of this problem where the sample $S$ and hypothesis $h$ are drawn from known distributions. Given a class $(X, H)$, let $D_X$ be a distribution over $X$, and $D_H$ a distribution over $H$. We say that $(D_X, D_H, X, H)$ is perfectly learnable in $q(n)$ expected queries if there exists an algorithm $A$ such that for every $n \in \mathbb{N}$, every sample $S \subset X$ of size $n$, and every hypothesis $h \in H$, $A$ correctly labels all of $S$ with respect to $h$ and uses at most $q(n)$ queries in expectation over $S \sim D_X$, $h \sim D_H$, and the internal randomness of the algorithm.

Perfect learning (or variants thereof) have long been known to share a close connection with active learning [13, 52, 3, 7]. In fact, a naive version of this relation is essentially immediate from definition—simply running a perfect learning algorithm on a sample of size $n = n(\varepsilon, \delta)$ results in an active PAC-learner with expected query complexity $q(n)$. In the next section, we'll cover this connection in slightly more depth.

## 2.5 Inference Dimension

In 2017, Kane, Lovett, Moran, and Zhang (KLMZ) [3] introduced *inference dimension*, a combinatorial parameter that exactly characterizes the query complexity of perfect learning under enriched queries. Inference dimension measures the smallest $k$ such that for all subsets $S$ of size $k$ and hypotheses $h \in H$, there always exists some $x \in S$ such that queries on $S \setminus \{x\}$ infer the label of $x$. Formally,

**Definition 2.2** (Inference Dimension). *The inference dimension of $(X, H)$ with query set $Q$ is the smallest $k$ such that for any subset $S \subset X$ of size $k$, $\forall h \in H$, $\exists x \in S$ s.t. $Q_h(S \setminus \{x\})$ infers $x$. If no such $k$ exists, then we say the inference dimension is $\infty$.*

KLMZ proved that query-efficient perfect learning is possible if and only if inference dimension is finite.

**Theorem 2.3** (Inference Dimension Characterizes Perfect Learning [3]). *Let $k$ denote the inference dimension of $(X, H)$ with respect to any binary query set $Q$, and let $Q(n)$ denote the worst-case number of queries required to learn the query response $Q(S)$ on any sample $S \subset X$ of size $n$. Then the expected query complexity of perfectly learning $(X, H)$ is:*

$$\Omega(\min(n, k)) \leq q(n) \leq O(Q(4k) \log n).$$

Furthermore, KLMZ prove as a corollary of this result that inference dimension also characterizes standard active learning.

**Theorem 2.4** (Inference Dimension Characterizes Active Learning [3]). *Let $(X, H)$ be a class with VC-dimension $d$ and inference dimension $k$ with respect to any query set $Q$. Then the query complexity of active learning $(X, H)$ is at most*[6]

$$q(\varepsilon, \delta) \leq O\left( Q(4k) \left( \log\left(\frac{d}{\varepsilon}\right) + \log\left(\frac{1}{\delta}\right) \right) \right).$$

*Furthermore, if $k = \infty$:*

$$q(\varepsilon, \delta) \geq \Omega(1/\varepsilon).$$

We note that when $Q$ is made up of label and derivative queries for a degree-$d$ PTF, $Q(4k) \leq O(dk)$. As a result of Theorem 2.3 and Theorem 2.4, the majority of our work analyzing worst-case models will focus on bounding the inference dimension. On a finer-grained level, it will also be useful to have an understanding of KLMZ's algorithm for classes with finite inference dimension, which is (roughly) given by the following basic boosting procedure:

**KLMZ Algorithm:** Denote the set of uninferred points at step $i$ by $X_i$.

1. Draw $4k$ points from $X_i$, and call this sample $S_i$.

2. Make all queries on $S_i$.

3. Remove all points in $X_i$ that can be inferred by $Q(S_i)$ to get $X_{i+1}$.

4. Repeat until $X_i$ is empty.

# 3 Worst-Case Active Learning PTFs

With background out of the way, we move to studying the query complexity of active learning PTFs in both the standard and batch worst-case models.

## 3.1 Classical Label Query Lower Bound

We'll start with a basic example showing that enriched queries are necessary for active learning PTFs. In fact, it turns out that even degree-two polynomials can't be efficiently active learned in the standard model. This follows from a well-known argument showing the same for the class of intervals on the real line.

**Lemma 3.1.** *The inference dimension of $(\mathbb{R}, H_2)$ is infinite with respect to label queries.*

---

[6]We note that this result does not appear as stated in [3], but follows immediately from their techniques.

*Proof.* It is enough to show there exists $h \in H_2$ and a subset $S \subset X$ with size $|S| = \infty$ such that no point can be inferred by other points in $S$. With this in mind, let $h(x) = x^2$ and set $S = \mathbb{N}$ to be all positive integers. Then $\text{sign}(h(x)) = +$ for all $x \in S$. However, we cannot infer any point $y \in S$ from $S \setminus \{y\}$ since we can find $g(x) = (x - y - \varepsilon)(x - y + \varepsilon)$ where $\varepsilon < \frac{1}{2}$ such that $\text{sign}(g(x)) = + = \text{sign}(h(x))$ for all $x \in S \setminus \{y\}$ but $\text{sign}(g(y)) \neq \text{sign}(h(y))$. $\qquad\square$

## 3.2 Upper Bounds with Derivative Queries

On the other hand, PTFs do have a very natural enriched query that admits query-efficient active learning: derivative queries. We'll show that the ability to query derivatives of all degrees[7] suffices to obtain an exponential improvement over standard passive query complexity bounds. In this section, we give two algorithms for efficiently active learning PTFs with derivative queries: a direct deterministic method through iterated binary search, and a randomized approach based on KLMZ's algorithm that extends nicely to the batch setting.

### 3.2.1 The Iterative Approach

We'll start by analyzing a basic iterative approach which gives the following characterization of perfect learning PTFs with derivative queries.

**Theorem 3.2** (Theorem 1.1, extended version). *The query complexity of active learning $(\mathbb{R}, H_d)$ with derivative queries is:*

$$\Omega(d \log n) \leq q(n) \leq O(d^3 \log n).$$

*Moreover, there is an algorithm achieving this upper bound that runs in $O(n(d + \log n))$ time.*

Proving Theorem 3.2 essentially boils down to arguing that we can use derivative information to easily identify monotone segments of any $f \in H_d$. Inference within each segment is then easy, as the restriction of $\text{sign}(f)$ on such a segment just looks like a threshold and can be learned by binary search. With this in mind, we break the proof of Theorem 3.2 into a couple of useful lemmas. First, we observe that it is possible to efficiently break any subset $S$ into a small number of segments sharing the same sign pattern.

**Lemma 3.3.** *For any degree-$k$ polynomial $f \in H_d$ and set $S = \{s_1 \leq \ldots \leq s_n\}$, given $\text{sign}(f^{(i)}(x))$ for all $1 \leq i \leq k$ and $x \in S$, it is possible to partition $S$ into $j \leq O(k^2)$ contiguous, disjoint segments*

$$I_1 = [s_1, s_{i_1}], \; I_2 = [s_{i_1+1}, s_{i_2}], \; \ldots, \; I_j = [s_{i_{j-1}+1}, s_n]$$

*such that each interval has a fixed sign pattern, i.e. for every $1 \leq \ell \leq j$ and $s, s' \in I_\ell$:*

$$\text{SgnPat}(s, f^{(1)}) = \text{SgnPat}(s', f^{(1)}).$$

*Moreover, this can be done in $O(n(k + \log n))$ time.*

*Proof.* Start by sorting the input set $S$. The intervals $I_i$ are defined by scanning through the sorted list and grouping together contiguous elements with the same sign pattern with respect to the first derivative, $\text{SgnPat}(g', \cdot)$. In other words, $i_j$ is given by the $j$th index such that $\text{SgnPat}(g', s_{i_j+1}) \neq \text{SgnPat}(g', s_{i_j})$.

We argue this process results in at most $O(k^2)$ total intervals. This follows from the intermediate value theorem, which promises that a root of some derivative must appear between each interval. More formally, observe that for any $1 \leq \ell < j$, we have by construction that the sign pattern of $g'$ changes between $s_{i_\ell}$ and $s_{i_\ell+1}$. By definition, this means some derivative must flip sign, and therefore crosses 0 somewhere in the interval $[s_{i_\ell}, s_{i_\ell+1}]$. On the other hand, the family of polynomials $\bigcup_{i=1}^{k-1} \{g^{(i)}\}$ has at most $k(k-1)/2$ total roots, so there cannot be more than $O(k^2)$ changes in sign pattern as desired.

$\qquad\square$

---

[7]Note that we actually do not need access to the $d_{th}$ derivative of $H_d$, which is always constant.

Second, we show that if two distinct points $a < b \in \mathbb{R}$ have the same sign pattern with respect to (the derivative of) $f \in H_d$, then $f$ is monotone on $[a, b]$.

**Lemma 3.4.** *Given a hypothesis $f \in H_d$, if $a < b \in \mathbb{R}$ satisfy $SgnPat(f', a) = SgnPat(f', b)$, then $f$ is monotone on $[a, b]$.*

*Proof.* We show that $f^{(i)}$ is monotone on $[a, b]$ for all $0 \le i \le d$ by reverse induction. This will suffice as the statement is precisely when $i = 0$.

When $i = d$, $f^{(i)}$ is a constant. For $0 \le i < d$, assume the result holds for degree $i + 1$. Since $a$ and $b$ have the same sign pattern on $f^{(i+1)}$ and $f^{(i+1)}$ is monotone on $[a, b]$ by the inductive hypothesis, we must be in one of the following two cases:

1. $f^{(i+1)} \ge 0$ on $[a, b]$. Then $f^{(i)}$ is non-decreasing on $[a, b]$.

2. $f^{(i+1)} \le 0$ on $[a, b]$. Then $f^{(i)}$ is non-increasing on $[a, b]$.

Thus $f^{(i)}$ is monotone in both possible cases so we are done. □

Thus the segments in Lemma 3.3 are monotone, and it is not hard to see that Theorem 3.2 is realized by the following basic procedure that iteratively learns each derivative starting from $f^{(d-1)}$:

1. Partition $f^{(i)}$ into $O((d-i)^2)$ monotone segments based on $SgnPat(f^{(i+1)}, S)$.

2. Run binary search independently on each segment

*Proof of Theorem 3.2.* We first prove the upper bound. To start, sort $S$ and learn the linear threshold function $\text{sign}(f^{(d-1)})$ by binary search. With this in hand, we can iteratively learn the $i$th derivative by the above process, as Lemma 3.4 promises we can divide each level into $(d-i)^2$ segments with fixed sign patterns given labels of all higher derivatives, and each segment is monotone by Lemma 3.4 so can be correctly labeled by binary search. At the end of this process we have learned the sign of all points in $S$ with respect to $f^{(0)}$ as desired. Finally, since we run at most $(d-i)^2$ instances of binary search in each iteration, the total process costs at most

$$\sum_{i=0}^{d-1} O((d-i)^2 \log n) \le O(d^3 \log n)$$

queries. The main computational cost comes from sorting $S$ and applying the scanning procedure in Lemma 3.4 $d$ times, for a total of $O(n(d + \log(n)))$ runtime.

The lower bound follows from a standard information theoretic argument: a set of $n$ points has at least $n^{\Omega(d)}$ possible labelings by degree $d$ polynomials, so we need at least $\log(n^{\Omega(d)}) = \Omega(d \log n)$ binary queries to solve the problem in expectation (and therefore also in the worst-case). □

### 3.2.2 Inference Dimension and Batch Active Learning

While the iterative approach gives a simple, deterministic technique for learning PTFs with derivative queries, it comes at the cost of a high amount of adaptivity. Even if one parallelizes the binary search at each level, the technique still requires $O(d \log(n))$ batch calls to the labeling oracle, and it is unclear whether the algorithm can be generalized to provide a trade-off between adaptivity and query complexity. In this section, we consider a simple algorithm based on KLMZ's inference dimension framework that overcomes this barrier via internal randomization, smoothly interpolating between the query-efficient and low-adaptivity regimes.

**Theorem 3.5.** *For any $n \in \mathbb{N}$ and $\alpha \in (1/\log(n), 1]$, there exists an algorithm for learning size $n$ subsets of $(\mathbb{R}, H_d)$ in*

$$q(n) \le O\left(\frac{d^3 n^\alpha}{\alpha}\right)$$

*expected queries, and*

$$r(n) \le 1 + \frac{2}{\alpha}$$

*expected rounds of adaptivity. Moreover, the algorithm can be implemented in $O(n/\alpha)$ time.*[8]

Note that when $\alpha = O(1/\log(n))$, Theorem 3.5 uses $O(d^3 \log(n))$ queries, matching the complexity of Theorem 3.2 (in expectation), but only requiring $O(\log(n))$ rounds of adaptivity. This is already a substantial improvement over the iterative approach as it is independent of degree, not to mention the broad freedom given in the generic choice of $\alpha$.

To prove Theorem 3.5, we rely on a simple extension of KLMZ's seminal work on inference dimension and active learning to the batch model.

**Theorem 3.6** (Inference Dimension $\to$ Batch Active Learning). *Let $(X, H)$ be a class with inference dimension $k$ with respect to query set $Q$. Then for any $n \in \mathbb{N}$ and $\alpha \in (1/\log(n), 1]$, there is an algorithm that labels any size $n$ subset of $X$ in*

$$q(n) \leq \frac{2Q_{total}(2kn^\alpha)}{\alpha}$$

*expected queries, and only*

$$r(n) \leq 1 + \frac{2}{\alpha}$$

*expected rounds of adaptivity, where $Q_{total}(m)$ is the total number of queries available on a set of $m$ points.*

We note the algorithm achieving Theorem 3.6 is essentially the standard algorithm given in Section 2.5, where the batch size $4k$ is replaced with $2kn^\alpha$. Plugging in $\alpha = 2/\log(n)$ recovers KLMZ's standard upper bound (Theorem 2.4). The proof of Theorem 3.6 follows from similar analysis to the original result [3, Theorem 3.2]. We include the proof in Appendix A for completeness.

Appealing to this framework, it is now enough to bound the inference dimension of $(\mathbb{R}, H_d)$ with respect to derivative queries. This follows from similar arguments to the technical analysis of our iterated approach. In particular, by Lemma 3.4 it is enough to show that any sample of $\Theta(d^2)$ points contains at least three with the same sign pattern, as such regions are monotonic and one point may always then be inferred.

**Lemma 3.7.** *Given a subsample $S \subset \mathbb{R}$ of size $|S| \geq d^2 + d + 3$ and any $f \in H_d$, there exist 3 consecutive points in $S$ (with respect to the natural ordering) that have the same sign pattern.*

*Proof.* By the same argument as Lemma 3.3, $S$ can be broken into $\frac{d(d+1)}{2} + 1$ segments where each segment has a fixed sign pattern with respect to $f$. The pigeonhole principle promises that if we $S$ has at least $d^2 + d + 3$ points then at least one of these segments must have at least 3 points, which share the same sign pattern by construction. □

Since $f$ is monotone on these segments, we get a bound on the inference dimension of $(\mathbb{R}, H_d)$.

**Corollary 3.8.** *The inference dimension of $(\mathbb{R}, H_d)$ with derivative queries is $O(d^2)$.*

*Proof.* Let $S \subset \mathbb{R}$ be any subsample of size $|S| \geq d^2 + d + 3$. By Lemma 3.7, for any $f \in H_d$, we know there exist at least three points (say $x_1, x_2, x_3$) of $S$ with the same sign pattern. By Lemma 3.4, $f$ is monotone on $[x_1, x_3]$, and since $\text{sign}(f(x_1)) = \text{sign}(f(x_3))$, $\text{sign}(f(x_2))$ can be inferred. □

Combining this with our batch variant of KLMZ gives the main result.

*Proof of Theorem 3.5.* The query and round complexity bounds follow immediately from combining Corollary 3.8 and Theorem 3.6. The analysis of computational complexity is slightly trickier. We'll assume $n \geq \text{poly}(d)$ for simplicity. The main expense lies in removing the set of inferred points in each round (sampling $O(d^2 n^\alpha)$ points to query from the remaining set takes sub-linear time in $n$ assuming access to uniformly random bits). We claim that removing the inferred points in each round can be done in linear time simply by scanning through $S$ and removing any points sandwiched between two queried points with the same sign pattern. We note that this departs slightly from the exact inference dimension algorithm of KLMZ which would use a linear program to infer all possible points. This algorithm corresponds to using

---
[8]We've assumed $n \geq \text{poly}(d)$ here for simplicity.

a 'restricted inference rule' that only infers within such monotone sections. A variant of KLMZ's algorithm for restricted inference is formalized in [30], and has the same guarantees. KLMZ's original algorithm can also be performed in polynomial time, but requires the points to have finite bit complexity which can be avoided with our argument. $\qquad\square$

## 3.3 Further Lower Bounds

We end the section by examining the tightness of our result in two additional senses:

1. Full access to derivatives is necessary: if we are missing *any* derivative, the inference dimension $k = \infty$.

2. Our inference dimension bound with respect to all derivatives is off by at most a factor of $d$:

$$\Omega(d) \leq k \leq O(d^2).$$

We'll start with the former. Let $Q_{\hat{i}}$ denote the query set containing all label and derivative queries with the exception of the $i$th derivative.

**Theorem 3.9.** *The inference dimension of $(\mathbb{R}, H_d)$ is infinite with respect to $Q_{\hat{i}}$ for any $1 \leq i \leq d-1$.*

*Proof.* We proceed by induction on the degree $d$. The base case is given by $(\mathbb{R}, H_2)$ where we are missing the first derivative. We remark that the construction from Section 3.1 still works in this case, since the second derivatives of $x^2$ and $(x - y + \varepsilon)(x + y - \varepsilon)$ are always $+$.

Now we perform the inductive step. We want to show the inference dimension of $(\mathbb{R}, H_d)$ is $\infty$ under $Q_{\hat{j}}$ for any $1 \leq j \leq d-1$. We'll break our analysis into two steps.

First, consider the case when $1 \leq j \leq d-2$. The induction hypothesis tells us for any $i < d$, $(\mathbb{R}, H_i)$ has inference dimension $\infty$ under $Q_{\hat{k}}$ for any $1 \leq k \leq i-1$. Since $j+1 \leq d-1$ we know by the induction hypothesis that $(\mathbb{R}, H_{j+1})$ has inference dimension $\infty$ for queries missing the $j_{th}$ derivative. That means there exists a degree $j+1$ polynomial $f$, an infinite set[9] $S \subset \mathbb{R}$ and a degree $j+1$ polynomial $f_t$ for each $s_t \in S$ such that

$$\operatorname{sign} f^{(k)}(s) = \operatorname{sign} f_t^{(k)}(s) \ \forall s \in S \backslash \{s_t\}, 1 \leq k \leq j-1.$$

Furthermore, since $f$ and $f_t^{(k)}$ are all degree $j+1$, the degree $k$ derivatives are trivial for $k > j+1$ and we have:

$$\operatorname{sign} f^{(k)}(s) = \operatorname{sign} f_t^{(k)}(s) \ \forall s \in S \backslash \{s_t\}, 1 \leq k \leq d-2, k \neq j,$$

which gives the desired result. This follows from the fact that when $k \in [j+1]\backslash\{j\}$, the statement is true by our construction, and when $k > j+1$, $f, f_k$ are both 0 (where $[n]$ denote $\{0, 1, \ldots, n\}$).

When $j = d-1$, we cannot reduce to lower degree and must provide a direct construction. Namely, we will construct a set $S \subset \mathbb{R}$ and a corresponding polynomial $h$ such that:

1. $|S| = \infty$.[10]

2. For any $s_i \in S$, there exists a degree $d$ polynomial $h_i$ such that

$$Q_{h_i}(S \setminus \{s_j\}) = Q_h(S \setminus \{s_j\})$$

for all $s_j \neq s_i, s_j \in S$, and $\operatorname{sign}(h_i(s_i)) \neq \operatorname{sign}(h(s_i))$.

This is sufficient to prove the result since it implies that for every $k \in \mathbb{N}$ there exists a set $S_k = \{s_1, \ldots, s_k\}$ and corresponding labeling $h$ such that no $s_i$ can be inferred by queries on the rest (since $h$ and $h_i$ are identical on all other points).

Let $h(x) = x^d$, so $h^{(j)} > 0$ on $(0, \infty)$ for all $j \in [d]$. We construct $S$ inductively. Given $s_1, \cdots, s_{n-1}$, we want to construct a polynomial $h_n$ and a point $s_n$ such that

---

[9]Note that infinite inference dimension does not strictly require such an infinite set, but it does hold for our particular induction.

[10]In particular $S$ is countably infinite.

1. $h_n^{(i)}(s_j) > 0 \ \forall j \in [n-1]\backslash\{0\}, \ \forall i \in [d-2] \cup \{d\}$.

2. $h_j^{(i)}(s_n) > 0 \ \forall j \in [n-1]\backslash\{0\}, \ \forall i \in [d-2] \cup \{d\}$.

3. $h_n(s_n) < 0$.

Define

$$h_n(x) = x^d - ds_{n-1}^3 x^{d-1} + d(d-1)s_{n-1}^4 x^{d-2}.$$

We claim that this $h_n$ satisfies these constraints when we pick $s_n = s_{n-1}^3 - 1$ recursively. To check this, note that for all $0 \leq i \leq d-2$ we have:

$$
\begin{aligned}
h_n^{(i)}(x) &= \frac{d!}{(d-i)!}x^{d-i} - ds_{n-1}^3 \frac{(d-1)!}{(d-1-i)!}x^{d-1-i} + d(d-1)s_{n-1}^4 \frac{(d-2)!}{(d-2-i)!}x^{d-2-i} \\
&= \frac{d!}{(d-i)!}x^{d-2-i}(x^2 - s_{n-1}^3(d-i)x + s_{n-1}^4(d-i)(d-i-1)) \\
&= \frac{d!}{(d-i)!}x^{d-2-i}f_{n,i}(x),
\end{aligned}
$$

where we define

$$f_{n,i}(x) = x^2 - s_{n-1}^3(d-i)x + s_{n-1}^4(d-i)(d-i-1).$$

For every $s_i > 0$, computing the sign of $h_n^{(i)}$ then reduces to analyzing $f_i$. We now show that the three conditions above hold, which completes the proof.

1. Consider $f_{n,i}(x)$. It is decreasing on $[0, \frac{s_{n-1}^3(d-i)}{2}]$, so for all $i = 0, \cdots, d-2$, $f_{n,i}(s_{n-1}) > 0$ will imply $f_{n,i}(s_j) > 0$ for all $j = 1, \cdots, n-1$. Checking $f_{n,i}(s_{n-1})$ directly gives

$$f_{n,i}(s_{n-1}) = s_{n-1}^2 - s_{n-1}^4(d-i) + s_{n-1}^4(d-i)(d-i-1) > 0$$

so we are done with this case.

2. Now consider $f_{\ell,i}(s_j)$ when $j > \ell$. We have:

$$f_{\ell,i}(s_j) = s_j^2 - s_{\ell-1}^3(d-i)s_j + s_{\ell-1}^4(d-i)(d-i-1).$$

Since $f_{\ell,i}(x)$ is increasing on $[\frac{s_{\ell-1}^3(d-i)}{2}, \infty)$ and $s_{\ell+1} = s_\ell^3 - 1 = (s_{\ell-1}^3 - 1)^3 - 1 > \frac{s_{\ell-1}^3(d-i)}{2}$, it suffices to check $f_{\ell,i}(s_{\ell+1}) > 0$:

$$
\begin{aligned}
f_{\ell,i}(s_{\ell+1}) &= s_{\ell+1}^2 - s_{\ell-1}^3(d-i)s_{\ell+1} + s_{\ell-1}^4(d-i)(d-i-1) \\
&= ((s_{\ell-1}^3 - 1)^3 - 1)((s_{\ell-1}^3 - 1)^3 - 1 - (d-i)s_{\ell-1}^3) + (d-i)(d-i-1)s_{\ell-1}^4 \\
&> 0.
\end{aligned}
$$

3. Finally, when $x = s_n$, by our construction we have $s_n = s_{n-1}^3 - 1$ so

$$
\begin{aligned}
f_{n,i}(s_n) &= (s_{n-1}^3 - 1)^2 - s_{n-1}^3(d-i)(s_{n-1}^3 - 1) + (d-i)(d-i-1)s_{n-1}^4 \\
&= (1 + i - d)s_{n-1}^6 + (d-i)(d-i-2)s_{n-1}^3 + (d-i)(d-i-1)s_{n-1}^4 + 1 \\
&< 0
\end{aligned}
$$

as long as $s_{n-1} > d!$. This condition can be satisfied by setting $s_1 = d!$. Since $\{s_n\}$ is strictly increasing, this is then satisfied for all $s_i$ including $s_{n-1}$.

$\square$

Finally, we close out the section by showing that even if derivatives of all degrees are present, the inference dimension of $(\mathbb{R}, H_d)$ is at least $\Omega(d)$, leaving just a linear gap between our analysis and the potentially optimal bound.

**Lemma 3.10.** *The inference dimension of $(\mathbb{R}, H_d)$ with derivative queries is $\Omega(d)$.*

*Proof.* Let

$$h(x) = \prod_{i=1}^{d}(x - r_i)$$

where $r_i$ are distinct real numbers and $r_1 < \ldots < r_d < 0$. Let $s_i = r_i + \varepsilon$ for some $\varepsilon > 0$, and define $h_i(x) = h(x)/(x - r_i)$ for all $1 \le i \le d$ and $g_i(x) = h_i(x)(x - (r_i + 2\varepsilon))$.

We claim that if $\varepsilon$ is small enough, then no point in $S = \{s_1, \ldots, s_d\}$ can be inferred by queries on the rest. It is enough to show that for all $1 \le i \le d$, $g_i(x)$ satisfies

1. $\text{sign}(g_i(s_i)) \neq \text{sign}(h(s_i))$ for all $1 \le i \le d$.

2. $\text{sign}(g_i^{(k)}(s_j)) = \text{sign}(h^{(k)}(s_j))$ for all $1 \le j \le d, j \neq i, 0 \le k \le d$.

Condition 1 holds by construction of $g_i$ as $r_i < s_i < r_i + 2\varepsilon$ for all $1 \le i \le d$. Similarly condition 2 holds by construction when $k = 0$, as $s_j - r_i$ and $s_j - (r_i + 2\varepsilon)$ have the same sign when $\varepsilon < \frac{1}{3} \min_{1 \le i \le d-1} |r_i - r_{i+1}|$. It is left to show that condition 2 holds when $1 \le k \le d$. To see this, notice that

$$h^{(k)}(x) = h_i^{(k)}(x)(x - r_i) + k h_i^{(k-1)}(x)$$

and

$$g_i^{(k)}(x) = h_i^{(k)}(x)(x - (r_i + 2\varepsilon)) + k h_i^{(k-1)}(x),$$

so

$$h^{(k)}(x) - 2\varepsilon h_i^{(k)}(x) = g_i^{(k)}(x). \tag{1}$$

Consider the set $T$ of roots of $h^{(k)}$ for all $1 \le k \le d$, that is:

$$T := \{x \in \mathbb{R} : \exists 1 \le k \le d, h^{(k)}(x) = 0\}.$$

Let $r_i' \in T$ be such that $r_i' \neq r_i$ and $|r_i' - r_i| > 0$ is minimal ($r_i \notin T$ since $h$ has no double roots). By letting $\varepsilon < \frac{1}{3} \min_{1 \le i \le d} |r_i' - r_i|$, we ensure that $s_i \notin T$, and therefore that $h^{(k)}(s_i) \neq 0$. Furthermore, since $h^{(k)}(r_i) \neq 0$ (no double roots) and there are no elements of $T$ between $r_i$ and $s_i$, it must be the case that $h^{(k)} \neq 0$ on the entire interval $[r_i, s_i]$. Since $h^{(k)}$ is continuous, it is bounded on $[r_i, s_i]$ and we can define:

$$u = \min_{1 \le k \le d} \left| \inf_{x \in [r_i, s_i]} h^{(k)}(x) \right| > 0,$$

and

$$w = \max_{\substack{1 \le k \le d \\ 1 \le i \le d \\ 1 \le j \le d, j \neq i}} |h_i^{(k)}(s_j)| \ge 0.$$

If $w = 0$ then $h_i^{(k)}(s_j) = 0$ for all $i, j, k, i \neq j$ and

$$h^{(k)}(s_j) = g_i^{(k)}(s_j)$$

and we are done. Otherwise $w \neq 0$. Notice that $u$ gets bigger when $\varepsilon$ gets smaller, and we can let

$$u_0 = \min_{1 \le k \le d} \left| \inf_{x \in [r_i, r_i + \frac{1}{3}|r_i' - r_i|]} h^{(k)}(x) \right| \le u.$$

Also $h_i^{(k)}$ is bounded on $[r_i, r_i + |r_i' - r_i|]$, so $w$ is globally bounded when $\varepsilon < \frac{1}{3} \min_{1 \le i \le d} |r_i' - r_i|$. Let

$$w_0 = \max_{\substack{1 \le k \le d \\ 1 \le i \le d \\ x \in [r_i, r_i + \frac{1}{3}|r_i' - r_i|]}} |h_i^{(k)}(x)|,$$

and therefore $w \leq w_0$. Since $w_0$ and $u_0$ are independent of $\varepsilon$, this means we can set $\varepsilon < \min(\frac{u_0}{2w_0}, \frac{1}{3} \min_{1 \leq i \leq d} |r_i - r_i'|) \leq \min(\frac{u}{2w}, \frac{1}{3} \min_{1 \leq i \leq d} |r_i - r_i'|)$ such that

$$|h^{(k)}(s_j)| > |2\varepsilon h_i^{(k)}(x)|,$$

which combined with Equation (1) implies that $\text{sign}(h^{(k)}(s_j)) = \text{sign}(g_i^{(k)}(s_j))$ for all $i, j, k, i \neq j$ as desired. □

# 4   Average-Case Active Learning PTFs

While worst-case analysis is a powerful tool for guarding against adversarial situations, in practice it is often the case that our sample and underlying classifier are chosen more at random than adversarially. In this section we'll analyze an average-case model capturing this setting. Notably, we'll show that in several natural scenarios, derivative queries actually are not necessary to achieve query efficient active learning. This is better suited than our worst-case analysis to practical scenarios like learning natural 3D-imagery, where we expect objects to come from nice distributions but don't necessarily have higher order information like derivatives.

To start, let's recall the specification of our model to the class of univariate PTFs $(\mathbb{R}, H_d)$: the learner is additionally given a distribution $D_X$ over $\mathbb{R}$ and $D_H$ over $H_d$. We are interested in analyzing the expected number of queries the learner needs to infer all labels of a sample drawn from $D_X$ with respect to a PTF drawn from $D_H$. Throughout this section, we will usually work with expectations over both $D_X$ and $D_H$, but it will sometimes be convenient to work only over $D_X$. As such, we'll use $\mathbb{E}_{D_X, D_H}$ throughout to denote the former, and $\mathbb{E}_{D_X}$ the latter.

We now present a simple generic algorithm for this problem we call "Sample and Search."

**Sample and Search Algorithm:** For any $f \in H_d$, note that $f$ has at most $d$ distinct real roots $\{r_i\}_{i=1}^d$. For notational convenience we denote $r_0 = -\infty$ and $r_{d+1} = \infty$. We design an algorithm to infer the labels of all points in $S \subset \mathbb{R}, |S| = n$:

1. Query the label (sign) of points from $S$ uniformly at random until either:

   (a) We have queried all $n$ points in $S$.

   (b) We see $d$ sign flips in the queried points, i.e. we have queried $x_1, \ldots, x_k$ and there exists indices $i_1 < \ldots < i_{d+1}$ such that
   $$\text{sign}(f(x_{i_j})) \neq \text{sign}(f(x_{i_{j+1}}))$$
   for all $j = 1, \ldots, d$.

2. If (b) occurred in the previous step, perform binary search on the points in $S$ between each pair $(x_{i_j}, x_{i_{j+1}})$ to find the sign threshold (and thereby labels) in that interval.

We note that a variant of this algorithm for a single interval (quadratic) is also discussed in [48], who note it can be used to achieve exponential rates in the active setting over any *fixed* choice of distribution and classifier.

We now argue Sample and Search correctly labels all points in $S$.

**Lemma 4.1.** *Sample and Search infers all labels of points in $S$.*

*Proof.* If we fall into 1(a), then trivially we know all the labels of $x \in S$. Otherwise we go to 1(b). Since $f$ is a degree $d$ polynomial, we can at most observe $d$ sign flips, and seeing exactly $d$ specifies a unique interval for every root. Thus performing a binary search on each interval returns the pair of points that are closest to each root of $f$, which is sufficient to infer the remaining points. □

Analyzing the query complexity of Sample and Search is a bit more involved. To answer this question, we will restrict our attention to distributions over PTFs with exactly $d$ real roots, though we note it is possible to handle more general scenarios query-efficiently via basic variants of Sample and Search if one is willing to move away from the perfect learning model.[11] In particular, as long as the polynomial family in question has sufficiently anti-concentrated roots, one can change the cut-off criterion in step 1 of Sample and Search to having drawn a sufficient number of random points to see each sign flip with high probability. This then incurs some small probability of error, which is allowed in the active model. Unfortunately, this technique cannot be used in the perfect learning model, which requires much more careful analysis due to its requirement of zero error.

To start our analysis, observe that the "Search" step of Sample and Search uses at most $d \log n$ queries, as it performs $d$ instances of binary search, so the main challenge lies in analyzing step 1. This is similar to the classical coupon collector problem, in which a collector draws from a discrete distribution over coupons until they have collected one of each type. In our setting, the "coupons" are made up by the intervals between adjacent roots,[12] and their probability is given by the mass of the marginal distribution on that interval. With this in mind, let $Y$ be the random variable measuring the number of samples required to hit each interval at least once, and let $Z = \min(Y, n)$.

**Proposition 4.2.** *The expected query complexity of the Sample and Search Algorithm is at most:*

$$q(n) \leq \mathbb{E}_{D_X, D_H}[Z] + d \log n.$$

*Proof.* Notice that $Z$ is exactly the variable measuring the number of queries used in step 1 by construction, so $\mathbb{E}_{D_X, D_H}[Z]$ is the expected number of queries needed in this step. Step 2 requires $d$ instances of binary search, so by linearity of expectation the expected query complexity of Sample and Search is at most $\mathbb{E}_{D_X, D_H}[Z] + d \log n$. □

It is worth noting that $\mathbb{E}_{D_X, D_H}[Z]$ and $\mathbb{E}_{D_X, D_H}[Y]$ can differ drastically. As a basic example, consider the case where $d = 1$ and we draw our $n$ points and one root uniformly at random from $[0, 1]$. It is a simple exercise to show that $\mathbb{E}_{D_X, D_H}[Y] = \infty$, whereas $\mathbb{E}_{D_X, D_H}[Z] = O(\log n)$ in this setting.

In the remainder of this section, we restrict our focus to working over $[0, 1] \subset \mathbb{R}$. In particular, both $S$ and the roots of $f \in H_d$ will be drawn from $[0, 1]$, and the former will always be chosen uniformly at random. We consider two potential distributions over the roots: the uniform and Dirichlet distributions.

## 4.1 Uniform distribution

We start by considering the uniform distribution over both points and roots. Let $U_{[0,1]}$ denote the uniform distribution on $[0, 1]$. We abuse notation to let $D_H = U_{[0,1]}$ also denote the distribution over $H_d$ where $d$ roots are chosen uniformly at random from $[0, 1]$.

**Theorem 4.3** (Theorem 1.4, extended version). $(U_{[0,1]}, U_{[0,1]}, \mathbb{R}, H_d)$ *is perfectly learnable with expected number of queries*

$$\Omega(d \log n) \leq q(n) \leq O(d^2 \log d \log n).$$

Most of the work in proving the upper bound in Theorem 4.3 lies in analyzing the random variable $Z$. To this end, we'll start with a basic lemma bounding the related variable $Y$ via standard analysis for the coupon collector problem.

**Lemma 4.4.** *For any $x \in \mathbb{R}_+$, let $E_x$ denote the event that $f \sim D_H$ has measure at least $\frac{1}{x}$ over $D_X$ between any two adjacent roots, the leftmost root and 0, and the rightmost root and 1. Then:*

$$\mathbb{E}_{D_X, D_H}[Y | E_x] \leq O(x \log d).$$

---

[11] While the perfect and active models are equivalent in the worst-case regime, it is not clear this is true in average-case settings.

[12] Note that this also includes the intervals $(r_0, r_1)$ and $[r_d, r_{d+1})$, where we recall $r_0 = -\infty$ and $r_{d+1} = \infty$

*Proof.* Let $Y_i$ denote the number of queries required to fill $i$ intervals after $i-1$ intervals have already been filled. Then

$$\mathbb{E}_{D_X, D_H}[Y|E_x] = \sum_{i=1}^{d+1} \mathbb{E}_{D_X, D_H}[Y_i|E_x].$$

Notice that

$$\mathbb{E}_{D_X, D_H}[Y_i|E_x] \leq \frac{x}{d+2-i},$$

because the probability of obtaining one of the $(d+1)-(i-1) = d+2-i$ intervals we are yet to collect is at least $\frac{d+2-i}{x}$. Therefore

$$\mathbb{E}_{D_X, D_H}[Y|E_x] \leq \sum_{i=1}^{d+1} \frac{x}{d+2-i} \leq O(x \log d)$$

by standard asymptotic bounds on the harmonic numbers. $\qquad\square$

Since $Z$ is just a cut-off of $Y$, we can use this fact combined with a bound on the minimum interval size to analyze the query complexity of Sample and Search.

**Proposition 4.5** (Upper bound). $\mathbb{E}_{D_X, D_H}[Z] \leq O(d^2 \log d \log n)$.

*Proof.* Let $M$ be the random variable giving minimal distance between any two adjacent root intervals, first root to 0, and last root to 1. By Lemma 4.4, we know $M \geq \frac{\log d}{x}$ implies $\mathbb{E}_{D_X, D_H}[Y|M] \leq O(x)$, and thus

$$\mathbb{P}_{D_H}[\mathbb{E}_{D_X}[Y] \leq x] \geq \mathbb{P}\left[M \geq \frac{c \log d}{x}\right] = \left(1 - \frac{c(d-1)\log d}{x}\right)^d$$

for any $x \in \mathbb{R}$ for some positive constant $c$.

Recall $Z = \min(Y, n)$ and $Z \geq d+1$, so $\mathbb{P}_{D_H}[\mathbb{E}_{D_X}[Z] \leq x] = 0$ when $x \leq d+1$. We can compute the expectation of $Z$ directly as:

$$\begin{aligned}
\mathbb{E}_{D_X, D_H}[Z] &= \int_0^\infty 1 - \mathbb{P}_{D_H}[\mathbb{E}_{D_X}[Z] \leq x] dx \\
&= \int_0^n 1 - \mathbb{P}_{D_H}[\mathbb{E}_{D_X}[Z] \leq x] dx \\
&= (d+1) + \int_{d+1}^n 1 - \mathbb{P}_{D_H}[\mathbb{E}_{D_X}[Z] \leq x] dx \\
&\leq (d+1) + \int_{d+1}^n 1 - \mathbb{P}_{D_H}[\mathbb{E}_{D_X}[Y] \leq x] dx \\
&\leq (d+1) + \int_{d+1}^n 1 - \mathbb{P}_{D_H}\left[M \geq \frac{c \log d}{x}\right] dx \\
&= (d+1) + \int_d^n 1 - \left(1 - \frac{c(d-1)\log d}{x}\right)^d dx \\
&\leq (d+1) + \int_d^n \frac{cd(d-1)\log d}{x} dx \\
&= (d+1) + cd(d-1)\log d(\log n - \log d) \leq O(d^2 \log d \log n)
\end{aligned}$$

$\qquad\square$

For the lower bound, we use classic information-theoretic arguments to show the standard worst-case bound continues to hold.

**Proposition 4.6.** $(U_{[0,1]}, U_{[0,1]}, \mathbb{R}, H_d)$ *requires at least*

$$q(n) \geq \Omega(d \log n)$$

*expected queries to perfectly learn, given* $n \geq \Omega(d^2)$.

*Proof.* We appeal to standard information theoretic arguments. In particular, notice that our problem can be rephrased as learning a binary string $L = \{\ell_1, \ldots, \ell_n\} \sim \{0, 1\}^n$ drawn from a known distribution $\mu$ via query access to the coordinates of $L$. This follows from the fact that each sample $S \sim [0, 1]^n$ and $h \in H_d$ corresponds to a particular pattern of labels, and therefore induces a fixed distribution $\mu$ over $\{0, 1\}^n$. With this in mind, notice that since our queries only give one bit of information, the expected number required to identify a sample $L$ from $\mu$ is at least the entropy $H(\mu)$.

It is left to argue that the $H(\mu) \geq \Omega(d \log n)$ for our particular choice of sample and hypothesis distributions. To see this, recall that our labeling is given by drawing a uniformly random sample of $n$ points from $[0, 1]$ and $d$ additional random roots from $[0, 1]$. By symmetry, this can be equivalently viewed as drawing $n + d$ points from $[0, 1]$ uniformly at random, and then selecting $d$ at random to be roots which results in $\binom{n+d}{d}$ equally distributed labelings. Denote this set of labelings as $\mathcal{L}$, then we can bound the entropy as

$$
\begin{aligned}
H(\mu) &= \sum_{L \in \mathcal{L}} \mathbb{P}[L] \log(1/\mathbb{P}[L]) \\
&= \sum_{L \in \mathcal{L}} \frac{1}{\binom{n+d}{d}} \log \binom{n+d}{d} \\
&= \log \binom{n+d}{d} \\
&\geq cd \log n
\end{aligned}
$$

for some universal constant $c > 0$. $\qquad\square$

## 4.2 Symmetric Dirichlet Distribution

In this section, we'll analyze another natural distribution over roots: the symmetric Dirichlet distribution (a generalization of choosing uniformly random points from a simplex). The Dirichlet distribution of order $m \geq 2$ with parameters $\alpha_1, \ldots, \alpha_m > 0$ has a probability density function

$$f(x_1, \ldots, x_m; \alpha_1, \ldots, \alpha_m) = \frac{1}{B(\boldsymbol{\alpha})} \prod_{i=1}^{m} x_i^{\alpha_i - 1}$$

where the support is over non-negative $x_i$ such that $\sum_{i=1}^{m} x_i = 1$, $\boldsymbol{\alpha} = (\alpha_1, \ldots, \alpha_m)$, the Beta function $B(\boldsymbol{\alpha})$ is the normalizing function given by

$$B(\boldsymbol{\alpha}) = \frac{\prod_{i=1}^{m} \Gamma(\alpha_i)}{\Gamma(\sum_{i=1}^{m} \alpha_i)},$$

and the Gamma function $\Gamma$ is defined as

$$\Gamma(z) = \int_{0}^{\infty} x^{z-1} e^{-x} dx.$$

We call the distribution *symmetric* if $\alpha_i = \alpha$ for all $1 \leq i \leq m$.

We consider the setting where the intervals between adjacent roots of $f \in H_d$ (along with 0 and 1) follow the symmetric Dirichlet distribution of order $d + 1$ with parameter $\alpha$, which we denote by

$$(x_1, \ldots, x_{d+1}) \sim \text{Dir}(\alpha).$$

In our analysis, it will often be useful to work over the marginal distribution of a given $x_i$. In this case, the marginals are given by the *Beta Distribution*, which with parameters $\alpha, \beta$ has probability density function

$$\frac{1}{B(\alpha, \beta)} x^{\alpha-1}(1-x)^{\beta-1},$$

where

$$B(\alpha, \beta) = \int_0^1 x^{\alpha-1}(1-x)^{\beta-1}dx.$$

In more detail, the marginal distribution of $\text{Dir}(\alpha)$ is a Beta distribution with parameters $\alpha_i, \sum_{\substack{j=1 \\ j\neq i}}^{d+1} \alpha_j$:

$$x_i \sim B(\alpha_i, \sum_{\substack{j=1 \\ j\neq i}}^{d+1} \alpha_j).$$

This simplifies to

$$x_i \sim B(\alpha, d\alpha)$$

in the symmetric case.

### 4.2.1 Upper Bound

In this section, we analyze the query complexity of $(U_{[0,1]}, \text{Dir}(\alpha), \mathbb{R}, H_d)$ for a few natural choices of $\alpha$.

**Theorem 4.7** (Theorem 1.5, extended version). *The query complexity of perfect learning $(\mathbb{R}, H_d)$ when the subsample $S \sim U_{[0,1]}$ and $h \sim Dir(\alpha)$ is at most*

$$q(n) \leq O(d^2 \log d \log n)$$

*when $\alpha = 1$,*

$$q(n) \leq O(d^2 \log d + d \log n)$$

*when $\alpha \geq 2$, and*

$$q(n) \leq O(d \log n)$$

*when $\alpha \geq d\log^2(n)$.*

Before proving these results, it is useful to prove the following general lemma on the form of $\mathbb{E}[Z]$.

**Lemma 4.8.** *In the setting $(U_{[0,1]}, Dir(\alpha), \mathbb{R}, H_d)$, we can upper bound $\mathbb{E}_{D_X, D_H}[Z]$ by*

$$\mathbb{E}_{D_X, D_H}[Z] \leq (d+1) + (d+1) \int_{d+1}^n \left( \frac{\int_0^{\frac{c \log d}{y}} x^{\alpha-1}(1-x)^{d\alpha-1}dx}{B(\alpha, d\alpha)} \right) dy$$

*where $c$ is the universal constant given in Proposition 4.5.*

*Proof.* Let $M = \min(x_i)$. We know

$$\mathbb{P}_{D_H}[\mathbb{E}_{D_X}[Z] \leq y] = 0$$

when $0 < y < d+1$. When $y \geq d+1$ we have by Lemma 4.4 that:

$$\mathbb{P}_{D_H}[\mathbb{E}_{D_X}[Z] \leq O(y)] \geq \mathbb{P}_{D_H}[M \geq \frac{\log d}{y}],$$

and therefore that

$$\mathbb{P}_{D_H}[\mathbb{E}_{D_X}[Z] \leq y] \geq \mathbb{P}_{D_H}[M \geq \frac{c \log d}{y}]$$

for some constant $c > 0$. Expanding out the expectation of $Z$ then gives:

$$\mathbb{E}_{D_X, D_H}[Z] = \int_0^n 1 - \mathbb{P}_{D_H}[\mathbb{E}_{D_X}[Z] \leq y] dy$$

$$= \int_0^{d+1} 1 - \mathbb{P}_{D_H}[\mathbb{E}_{D_X}[Z] \leq y] dy + \int_{d+1}^n 1 - \mathbb{P}_{D_H}[\mathbb{E}_{D_X}[Z] \leq y] dy$$

$$\leq (d+1) + \int_{d+1}^n 1 - \mathbb{P}_{D_H}[M \geq \frac{c \log d}{y}] dy$$

$$= (d+1) + \int_{d+1}^n \mathbb{P}_{D_H}[M \leq \frac{c \log d}{y}] dy$$

$$\leq (d+1) + (d+1) \int_{d+1}^n \mathbb{P}_{D_H}[x_1 \leq \frac{c \log d}{y}] dy$$

$$= (d+1) + (d+1) \int_{d+1}^n \left( \frac{\int_0^{\frac{c \log d}{y}} x^{\alpha-1}(1-x)^{d\alpha-1} dx}{B(\alpha, d\alpha)} \right) dy$$

where the second-to-last inequality comes from a union bound:

$$\mathbb{P}_{D_H}[M \leq \frac{c \log d}{y}] \leq \sum_{i=1}^{d+1} \mathbb{P}_{D_H}[x_i \leq \frac{c \log d}{y}] \leq (d+1) \mathbb{P}_{D_H}[x_1 \leq \frac{c \log d}{y}].$$

$\square$

With this in mind, we'll now take a look at the setting where $\alpha = 1$, called the "flat Dirichlet distribution." This corresponds to choosing a uniformly random element on the $d$-simplex.

**Lemma 4.9.** *In the setting* $(U_{[0,1]}, Dir(1), \mathbb{R}, H_d)$,

$$\mathbb{E}_{D_X, D_H}[Z] \leq O(d^2 \log d \cdot \log n).$$

*Proof.* We continue our computation in Lemma 4.8 with $\alpha = 1$:

$$\mathbb{E}_{D_X, D_H}[Z] \leq (d+1) + \frac{d+1}{B(1,d)} \int_{d+1}^n \int_0^{\frac{c \log d}{y}} (1-x)^{d-1} dx dy$$

$$= (d+1) + d(d+1) \int_{d+1}^n \left( \frac{1}{d} - \frac{(1 - \frac{c \log d}{y})^d}{d} \right) dy$$

$$= (d+1) + (d+1) \int_{d+1}^n 1 - \left( 1 - \frac{c \log d}{y} \right)^d dy$$

$$\leq (d+1) + (d+1) \int_{d+1}^n \frac{cd \log d}{y} dy$$

$$= (d+1) + cd(d+1) \log d \cdot (\log n - \log(d+1))$$

$$\leq O(d^2 \log d \cdot \log n).$$

$\square$

As an immediate corollary, we get the desired upper bound on the query complexity of $(U_{[0,1]}, \mathrm{Dir}(1), \mathbb{R}, H_d)$.

**Corollary 4.10.** $(U_{[0,1]}, Dir(1), \mathbb{R}, H_d)$ *is perfectly learnable in at most*

$$q(n) \leq O(d^2 \log d \cdot \log n)$$

*expected queries.*

On the other hand, a more careful analysis shows that the upper bound improves non-trivially as $\alpha$ grows. First, we show that when $\alpha = 2$, the query complexity is at most $\tilde{O}(d^2 + d\log n)$, which can be bounded by $O(d\log n)$ when $n$ is sufficiently large.

**Lemma 4.11.** *In the setting* $(U_{[0,1]}, Dir(2), \mathbb{R}, H_d)$,

$$\mathbb{E}_{D_X, D_H}[Z] \leq O(d^2 \log d)$$

*Proof.* We continue our computation in Lemma 4.8 with $\alpha = 2$:

$$\mathbb{E}_{D_X, D_H}[Z] \leq (d+1) + (d+1)\int_{d+1}^{n} \mathbb{P}_{D_H}[x_1 \leq \frac{c\log d}{y}]dy$$

A change of variable with $z = \frac{y}{c\log d}$ gives us[13]

$$\mathbb{E}_{D_X, D_H}[Z] \leq (d+1) + c(d+1)\log d \int_{\frac{d+1}{c\log d}}^{\frac{n}{c\log d}} \mathbb{P}_{D_H}[x_1 \leq \frac{1}{z}]dz$$

$$\leq (d+1) + c(d+1)\log d \left( \int_0^1 \mathbb{P}_{D_H}[x_1 \leq \frac{1}{z}]dz + \int_1^\infty \mathbb{P}_{D_H}[x_1 \leq \frac{1}{z}]dz \right)$$

$$= c'd\log d + \frac{c(d+1)\log d}{B(2,2d)} \int_1^\infty \left( \int_0^{1/z} x(1-x)^{2d-1}dx \right) dz$$

$$= c'd\log d + \frac{c(d+1)\log d}{B(2,2d)} \int_1^\infty \frac{1}{2d(2d+1)} - \frac{(1-1/z)^{2d}(\frac{2d}{z}+1)}{2d(2d+1)}dz$$

$$= c'd\log d + c(d+1)\log d \int_1^\infty 1 - (z-1)^{2d}\left(\frac{2d}{z}+1\right)z^{-2d}dz$$

where $c' > 0$ is some universal constant. It remains to compute the integral, which can be checked directly by noting the anti-derivative of the integrand is $z - (z-1)\left(\frac{z-1}{z}\right)^{2d}$:

$$\int_1^\infty 1 - (z-1)^{2d}(\frac{2d}{z}+1)z^{-2d}dz = \lim_{z\to\infty} z - (z-1)\left(\frac{z-1}{z}\right)^{2d} - 1 = 2d.$$

Plugging this into the above gives:

$$\mathbb{E}_{D_X, D_H}[Z] \leq c'd\log d + 2cd(d+1)\log d$$
$$\leq O(d^2 \log d)$$

as desired. $\square$

As an immediate corollary, we get the following bound on the query complexity of $(U_{[0,1]}, Dir(2), \mathbb{R}, H_d)$.

**Corollary 4.12.** $(U_{[0,1]}, Dir(2), \mathbb{R}, H_d)$ *is perfectly learnable in at most*

$$q(n) \leq O(d^2 \log d + d\log n)$$

*expected queries.*

When $\log n \geq \Omega(d\log d)$, i.e. $n \geq \Omega(d^d)$, note that the above becomes $O(d\log n)$. We will show this bound is tight in the next section.

Finally, we'll show that as we take $\alpha$ sufficiently large, the extraneous $d^2 \log(d)$ term disappears.

---

[13]We note that we are abusing notation a bit for simplicity in the second equation below. The integral does not actually need to go to 0 (and thus there is no issue with the $1/z$ in the integrand).

**Theorem 4.13.** $(U_{[0,1]}, Dir(\alpha), \mathbb{R}, H_d)$ *is perfectly learnable in at most*

$$q(n) \leq O(d \log n)$$

*expected queries when* $\alpha \geq \Omega(\log^2 n)$.

*Proof.* Let $M = \min(x_i)$. By Lemma 4.4 we know that when $M \geq \frac{1}{2d}$, then $\mathbb{E}_{D_X, D_H}[Z] \leq O(d \log d)$. Therefore, we can break up $\mathbb{E}_{D_X, D_H}[Z]$ into two parts based on $M$:

$$\mathbb{E}_{D_X, D_H}[Z] \leq \mathbb{P}_{D_H}[M \geq \frac{1}{2d}]O(d \log d) + \mathbb{P}_{D_H}[M < \frac{1}{2d}]n \tag{2}$$
$$\leq O(d \log d) + \mathbb{P}_{D_H}[M < \frac{1}{2d}]n.$$

Recall that the marginal distribution of $\mathrm{Dir}(\alpha)$ is the Beta distribution $B(\alpha, d\alpha)$. The tail behavior of the Beta distribution is well-understood: as $\alpha$ grows large $B(\alpha, d\alpha)$ becomes increasingly concentrated around its expectation. In particular, appealing to concentration bounds in [53, Theorem 1] with $\mathbb{E}[B(\alpha, d\alpha)] = \frac{1}{d+1}$, we have

$$\mathbb{P}_{D_H}[M < \frac{1}{2d}] \leq (d+1)\mathbb{P}_{D_H}[x_1 < \frac{1}{2d}]$$
$$\leq (d+1)\mathbb{P}_{D_H}\left[\left|x_1 - \frac{1}{d+1}\right| > \frac{1}{d+1} - \frac{1}{2d}\right]$$
$$\leq (d+1)e^{-c\alpha^{1/2}},$$

for some universal constant $c > 0$.[14] Taking $\alpha \geq \log^2(n)/c$ then gives:

$$\mathbb{P}_{D_H}[M < \frac{1}{2d}] \leq O(d/n).$$

Plugging this result into Equation (2) gives

$$\mathbb{E}_{D_X, D_H}[Z] \leq O(d \log d),$$

and combining this fact with Proposition 4.2 results in the desired query complexity of

$$q(n) \leq O(d \log n).$$

$\square$

### 4.2.2 Lower Bound

We'll close the section with the query lower bounds for perfectly learning $(U_{[0,1]}, \mathrm{Dir}(\alpha), \mathbb{R}, H_d)$. Namely, we show that the same result as the worst and uniform cases continues to hold, albeit with some dependence on $\alpha$.

**Proposition 4.14.** *The expected query complexity of perfectly learning* $(U_{[0,1]}, Dir(\alpha), \mathbb{R}, H_d)$ *is at least*

$$q(n) \geq \Omega_\alpha(d \log n),$$

*where we have suppressed dependence on* $\alpha$.

*Proof.* The same method used in Proposition 4.6 can be applied here: it is sufficient to show that the entropy of $\mathrm{Dir}(\alpha)$ discretized to the uniformly random point set $S$ is $\Omega_\alpha(d \log n)$. The trick is to notice that this is exactly the well-studied "Dirichlet-Multinomial" distribution whose asymptotic entropy is known [54, Theorem 2]:

$$H(\mu) = (d-1) \log n - O_\alpha(1) - o_n(1).$$

This gives the desired result. $\square$

---

[14]We note this is not the exact form that appears in [53], but it follows without much difficulty from plugging in our parameter setting.

By our previous analysis, this implies Sample and Search is optimal for constant $\alpha \geq 2$ when $n$ is sufficiently large (we only show $\alpha = 2$, but the algorithm performance only improves as $\alpha$ increases).

# 5  Beyond Univariate PTFs

In this section, we show that derivative queries are insufficient to learn multivariate PTFs. In particular, we show that the inference dimension of $(\mathbb{R}^2, H_2^2)$ is infinite even when the learner has access to the gradient and Hessian, where $H_2^2$ is the class of two-variate quadratics. More formally, we consider a learner which can make label queries, *gradient queries* of the form $\mathrm{sign}\left(\frac{\partial f}{\partial x}(x_1, y_1), \frac{\partial f}{\partial y}(x_1, y_1)\right)$, and *Hessian queries* of the form $\mathrm{sign}\left(\frac{\partial^2 f}{\partial x \partial x}(x_1, y_1), \frac{\partial^2 f}{\partial x \partial y}(x_1, y_1), \frac{\partial^2 f}{\partial y \partial x}(x_1, y_1) \frac{\partial^2 f}{\partial y \partial y}(x_1, y_1)\right)$ for any $(x_1, y_1) \in \mathbb{R}^2$ in the learner's sample.

**Theorem 5.1.** *The inference dimension of $(\mathbb{R}^2, H_2^2)$ with label, gradient, and Hessian queries is infinite.*

*Proof.* Consider the set $S = \{(x_1, y_1), \cdots, (x_n, y_n)\}$ where $x_i = \sin(\frac{\pi}{2(n+1)}i + \frac{\pi}{2})$ and $y_i = \cos(\frac{\pi}{2(n+1)}i + \frac{\pi}{2})$ with $1 \leq i \leq n$ and two functions $h(x) = -x^2 - y^2 - \epsilon xy$ and $h'(x) = -x^2 - y^2 + \epsilon xy$ where $\epsilon \leq \min_{(x,y) \in S}(\min(|\frac{x}{y}|, |\frac{y}{x}|))$. Note that the value of $h$ and $h'$, their partial derivatives, and the diagonal elements of Hessians evaluated on $S$ are all negative. The off-diagonal entries of the Hessian are all positive on $h'$ and negative on $h$. We claim that we cannot infer any point $s_i$ from $S \backslash \{s_i\}$ no matter the size of $n$. To show this, it is enough to construct a hypothesis having same label, gradient, and Hessian queries on all the points in $S$ with either $h$ or $h'$ except being positive on $s_i$.

To this end, for each $1 \leq i \leq n$ consider the hypothesis

$$h_i(x, y) = (x \cos(\theta_i) - y \sin(\theta_i))(x \sin(\theta_i) + y \cos(\theta_i)) - c_1(x \sin(\theta_i) + y \cos(\theta_i))^2 - c_2(x^2 + y^2 - 1),$$

where $\theta_i = -\frac{\pi}{4(n+1)} - \frac{\pi}{2(n+1)}(i - 1)$, $c_1 = \frac{1}{\tan(\frac{\pi}{2(n+1)})}$, and $c_2 = c_1^2 + c_1 + 1$.

Notice that $h_i(x, y)$ is the result of spinning the function $f(x, y) = xy - c_1 y^2$ counter-clockwise by $\theta_i$ radians and subtracting $c_2(x^2 + y^2 - 1)$. Since this last addition has no effect on the sign of points in $S$, we can determine the sign of $h_i$ on each $s_j$ by examining the sign and rotation of $f$. In particular, notice that the

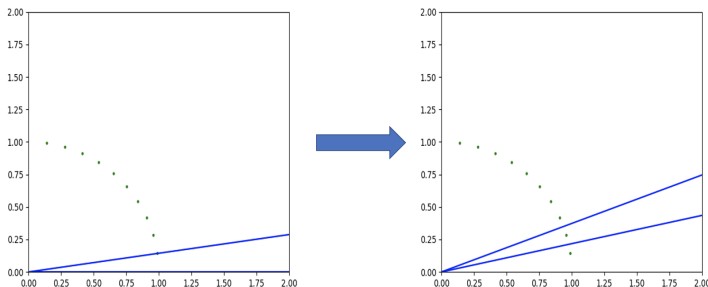

Figure 1: An example of spinning $f = xy - c_1 y^2$ by $|\theta_2| = \frac{\pi}{4(n+1)} + \frac{\pi}{2(n+1)}$ when $n = 10$. Notice that every point in the first quadrant is negative except the one between the blue lines.

value of $f(x)$ in the first and fourth quadrants is only positive between the lines $y = 0$ and $y = \frac{1}{c_1}x$, which make a circular sector with central angle $\arctan(1/c_1) \leq \frac{\pi}{2(n+1)}$. Since the points in $S$ are separated by $\frac{\pi}{2(n+1)}$ radians, it is clear that after rotation the positive sector only contains one point in $S$—namely that $h_i$ is positive on $s_i$, and negative on $s_j$ for all $j \neq i$.

It is left to show that the gradients and Hessian of $h_i$ remain negative for every $s_j$ where $j \neq i$, which we do

by direct computation. Namely we claim that the partial derivatives at $u_j = (x_j, y_j)$,

$$\frac{\partial h_i(u_j)}{\partial x} = (2\sin(\theta)\cos(\theta) - 2c_1\sin^2(\theta) - 2c_2)x_j + (\cos^2(\theta) - 2c_1\sin(\theta)\cos(\theta) - \sin^2(\theta))y_j$$

and

$$\frac{\partial h_i(u_j)}{\partial y} = (\cos^2(\theta) - 2c_1\sin(\theta)\cos(\theta) - \sin^2(\theta))x_j + (-2\sin(\theta)\cos(\theta) - 2c_1\cos^2(\theta) - 2c_2)y_j$$

are both negative. To see this, note that the ratios $\frac{x_j}{y_j}$ and $\frac{y_j}{x_j}$ are bounded: $x_j \leq \frac{1}{\tan(\frac{\pi}{2(n+1)}i)}y_j \leq c_1 y_j$ and $y_j \leq \frac{1}{\tan(\frac{\pi}{2(n+1)}i)}x_j \leq c_1 x_j$. This means that choosing $c_2$ large enough makes $-2c_2 x_j$ the dominant term in $\frac{\partial h_i(x_j, y_j)}{\partial x}$, and $-2c_2 y_j$ the dominant term in $\frac{\partial h_i(x_j, y_j)}{\partial x}$. It can be checked directly that setting $c_2 \geq c_1^2 + c_1 + 1$ is then sufficient to turn both partial derivatives negative. Similarly we can compute the Hessian:

$$Hessian(h_i(u_j)) = \begin{bmatrix} (2\sin(\theta)\cos(\theta) - 2c_1\sin^2(\theta) - 2c_2) & (\cos^2(\theta) - 2c_1\sin(\theta)\cos(\theta) - \sin^2(\theta)) \\ (\cos^2(\theta) - 2c_1\sin(\theta)\cos(\theta) - \sin^2(\theta)) & (-2\sin(\theta)\cos(\theta) - 2c_1\cos^2(\theta) - 2c_2) \end{bmatrix},$$

and observe that the diagonal entries are negative when evaluated on any point in $S$. Notice that the off-diagonal entries are same for each points in $S$. By the pigeonhole principle, at least half of points are either labeled 1 or $-1$ for off-diagonal entries of hessian. We will use $h$ if more than half are labeled $-1$ for off-diagonal entries, and $h'$ otherwise. $\qquad \square$

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

# A  Extending KLMZ to the Batch Model

In this section we give a batch-variant of KLMZ's seminal learning algorithm. The basic idea remains the same as in the algorithm discussed in Section 2.5. but the number of points drawn in step 1 of each iteration is generalized from $4k$ to a generic batch size $m$. For the sake of analysis, it is actually simpler to consider a slightly more complicated variant of this algorithm (indeed this is done in KLMZ as well). In this variant, we divide our algorithm into $\frac{\log(n)}{\log(\frac{m}{2k})}$ iterations, where in each iteration we aim to learn all but a $\frac{2k}{m}$ fraction of the remaining points. In particular, the $i$th iteration repeatedly draws samples of size $m$ from $X_i$ until the total number of un-inferred points is at most $n\left(\frac{2k}{m}\right)^i$.

---

**Algorithm 1:** BATCH-KLMZ$(S, m)$

> **Result:** Labels all points in $S$
> **Input:** Class $(X, H)$, Subset $S \subseteq X$, Query set $Q$, Query Oracle $O_Q$
> **Parameters:**
> - Inference dimension $k$
> - Batch size $m$
> - Iteration cutoff $t = \frac{\log(n)}{\log(\frac{m}{2k})}$
>
> **Algorithm:**
> $S_0 \leftarrow S$
> **for** $i$ *in range* $t$ **do**
> > $T \leftarrow \{\varnothing\}$
> > **while** $Cov_{S_i}(Q_h(T)) < \frac{m-2k}{m}$ **do**
> > > Sample $T \sim S_i^m$
> > > Query $O_Q(T)$
> >
> > **end**
> > $S_{i+1} \leftarrow \{x \in S_i : Q_h(T) \not\rightarrow_h x\}$
> > **if** $|S_{i+1}| \leq m$ **then**
> > > Query $O_Q(S_{i+1})$
> > > **Return**
> >
> > **end**
>
> **end**

---

We show that the round complexity of BATCH-KLMZ is at most $O(\frac{\log(n)}{\log(\frac{m}{2k})})$.

**Theorem A.1.** *Let $(X, H)$ be a class with inference dimension $k$ with respect to query set $Q$. Then for any $S \subseteq X$ satisfying $|S| = n$ and any $m > 2k$, BATCH-KLMZ$(S, m)$ correctly labels all points in $S$ using only*

$$r(n) = 1 + \frac{2\log(n)}{\log(\frac{m}{2k})}$$

*expected rounds of adaptivity, and*

$$q(n) = Q_{total}(m)r(n)$$

*queries in expectation, where $Q_{total}(m)$ is the total number of queries available on a set of $m$ points.*

Setting the batch size to $m = 2kn^\alpha$ gives the form of the result appearing in the main body.

The core proposition used to prove this result is a bound on the coverage of $m$ uniformly random points from $X$. This is analyzed for the setting $m = 4k + 1$ in KLMZ, but is easy to extend to the general setting by analogous arguments.

**Lemma A.2** ([3, Lemma 3.3]). *Let $(X, H)$ be a size $n$ class with inference dimension $k$. Then for any distribution $D$ over $X$ and $h \in H$, the coverage of $Q_h(S)$ over $S \sim D^m$ is large with constant probability:*

$$\Pr_{S \sim D^m} \left[ Cov(Q_h(S)) \geq \frac{m - 2k}{m} \right] \geq 1/2.$$

With this in hand, the proof of Theorem A.1 is essentially immediate from linearity of expectation.

*Proof of Theorem A.1.* Recall that the algorithm is performed in $t$ iterations, where the $i$th iteration is promised to contract the remaining number of un-inferred points by a factor of at least $\frac{2k}{m}$. Thus after $t = \frac{\log(n)}{\log(\frac{m}{2k})}$ iterations there can be at most $(2k/m)^t n = 1$ points remaining, and the algorithm therefore infers all points by the $(t + 1)$st round as desired.

It is left to analyze the expected number of batch oracle calls within each iteration. In particular, let $w_i$ be the random variable denoting the number of times the while statement loops in iteration $i$. By linearity of expectation, the expected number of rounds of adaptivity is then:

$$r(n) = \sum_{i=1}^{t} \mathbb{E}[w_i].$$

By Lemma A.2, the probability iteration $i$ terminates in any run of the loop is at least $1/2$, which implies $\mathbb{E}[t_i] \leq 2$ and gives the desired round complexity. The query complexity is immediate from the batch size $m$. $\square$