# OpenReview forum: "Active Learning Polynomial Threshold Functions"
_NeurIPS.cc/2022/Conference — NeurIPS 2022 Accept_

### Official Review · Reviewer_up2u · 2022-07-06

**Rating:** 7
**Confidence:** 3
**Soundness:** 3 good
**Presentation:** 4 excellent
**Contribution:** 3 good

**Summary:**

The submission provides theory and algorithms for active learning polynomial threshold functions under access to derivatives of the underlying classifier. The submission provides analysis of proposed algorithms that have desirable query complexity and rounds of adaptivity.

**Questions:**

Some minor questions are listed here:

- Theorem 1.6 suggests that derivative queries for multivariate PTFs are insufficient in the worst case setting. This doesn't seem to be too surprising considering the fact that missing one derivative is undesirable for univariate PTFs. Are there other types of additional query information for the multivariate case so that perfect learning is possible with small amounts of queries?
- There exists quite a gap between the lower and upper bounds in Theorem 1.1; is there any intuition between the gap and what may be causing it? For example, is the lower bound too loose, the upper bound too loose, or a bit of both?
- This is a nice theory work --- can the practical implications / limitations of this work be clarified even further? Particularly, in the *worst case*, if all derivatives (up to degree $d$) are required, it seems like this would be an issue and would not be practically feasible.

**Limitations:**

Some limitations are discussed in the paper. See above sections for suggestions/questions for improvement.

**Strengths And Weaknesses:**

The strengths and weaknesses of the submission are compiled below.

**Originality.** The paper looks at active learning polynomial threshold functions with access to the signs of derivatives of the underlying classifier, which is a new topic within the many previous related work. The paper uses some real estate to demonstrate that derivative queries are well motivated in theory and practice (line 45 and the corresponding paragraph) and the discovery that perfect learning under the worst case requires derivatives (Theorem 1.3) is quite interesting.

**Quality and clarity.** Overall, the paper is nicely written. Past history and relevant literature discussions are adequate. Some minor questions are attached to the next "Questions" section.

**Significance.** The paper is an interesting addition to a long list of papers in the active learning community and its topic and results are of adequate significance.

---

> ### Author Response · Authors · 2022-07-29
> **Rebuttal**
>
> We thank the reviewer for their insightful comments and questions.
>
> It is certainly true that in worst-case settings, a human oracle would likely find it difficult to answer derivative queries beyond second order. Indeed this is one of the main motivations of our study of the average-case setting, since we prove that higher order derivatives are unfortunately necessary in the worst-case. In this sense our average case results are rather encouraging, showing that derivatives are not needed at all for various practically reasonable distributions.
>
> On the other hand, there are settings where we believe derivative queries (even of higher order) are practically relevant, for example in experimental design applications where a learner may be able to accurately configure the experiment parameters.
>
> The questions about the multivariate case and about closing the gap between the upper and lower bounds are indeed intriguing open problems. See the joint part of our response (to all reviewers) for more details.

---

> > ### Comment · Reviewer_up2u · 2022-08-08
> > **Thanks for responses**
> >
> > Thanks for the detailed responses to my questions.

---

### Official Review · Reviewer_SBJT · 2022-07-11

**Rating:** 8
**Confidence:** 4
**Soundness:** 4 excellent
**Presentation:** 4 excellent
**Contribution:** 4 excellent

**Summary:**

This paper introduces the problem of active learning of univariate $d$-degree polynomial threshold functions using so-called derivative queries, which are a novel, yet, natural enriched query type for this learning problem.

The authors discuss that usual label queries are not sufficient to learn in this setting already for degree 2 polynomials (corresponding to the well-known problem in active learning that learning linear thresholds / non-homogeneous halfspaces already in R^2 is not possible).

In the perfect learning setting (inferring all labels of a possibly adversarially given finite unlabelled dataset) the authors state near-tight lower and upper bounds and devise a algorithm achieving the query upper bound with near-linear runtime.

These results for perfect learning immediately imply bounds for classical $(\epsilon,\delta)$-based (PAC) active learning. However, to generalise the result to average-case bounds a more involved analyses is required.

They also state some distribution-dependent result for common families of distributions.

Lastly, they propose a simple variant where the points are not queried one-by-one but rather in batches and give an upper bound in terms of the batch-size.

**Questions:**

There is still a gap in the inference dimension of the problem ($\Omega(d)$, and $O(d^2)$). Do the authors have first ideas or a hunch which one of the two is the "correct" tight one?

While the kernel/embedding view of this problem is shortly discussed, I would like to see a deeper discussion of this issue. How usual bounds for learning halfspaces (in the polynomial feature space) relate to the achieved here?

**Limitations:**

I don't thing that this work has limitations. Still some pointers and questions to look into:

Are large-margin bounds possible? Maybe related: Do bounds on the Lipschitz constant of the polynomial help?

Are there (simple) first positive results for multivariate polynomials? Maybe with even more powerful queries?

There are ways to completely get rid of the $\log 1/\delta$ in active learning upper bounds (Hanneke and Yang, 2015). Is this possible/useful here?

**Strengths And Weaknesses:**

This paper is a pleasant read and introduces a very interesting and quite natural active learning problem. The active learning community and general learning theory community should be interested in this work.

The authors discuss related work and introduce necessary background.

This work is a strong introduction into this novel learning problem, devising near-tight bounds and efficient algorithms. Also various related aspects are discussed.

Minor suggestions:
* is perfect learning the same as "exact learning" (e.g., Hegedűs, Tibor, COLT 1995). Maybe mention it, as well in the related work.
* $f^{(0)}$ is not really defined, is it just $f$ itself (see e.g., line 106).
* the $p^{(i)}$ notation seems to be not yet defined in line 60.
* line 344: a "," missing in the Hessian? Also how exactly is the sign of a vector defined?

---

> ### Author Response · Authors · 2022-07-29
> **Rebuttal**
>
> We thank the reviewer for the thorough review and insightful comments. We will fix the minor suggestions throughout. We comment briefly below on the reviewer's additional questions.
>
> The large margin case can be solved by embedding into higher dimensions and using comparison queries. It is not clear that margin would improve the inference dimension under derivative queries, since intuitively it does not help you `find roots.' That said, it is not impossible that combined with some smoothness condition the sign patterns are better behaved and might lead to smaller inference dimension.
>
> Removing the $\delta$ dependence is often a tricky question when enriched queries are involved. While this can be done in standard active learning via the splitting-index/star-dimension, the only derandomized version of a related enriched query algorithm is given in the paper "Generalized comparison trees for point-location problems."[KLM18]  However the method is tailored to comparison queries, loses polynomial factors in dimension, and does not immediately imply the removal of $\delta$ in the active setting (naively it would only improve the dependence to $\log\log(1/\delta)$).
>
> Closing the inference dimension gap and exploring the multivariate case (perhaps under more powerful queries) are among the most interesting open questions. We discuss them in detail in the joint response (to all reviewers).

---

> > ### Comment · Reviewer_SBJT · 2022-08-03
> > **Thanks for clarifying**
> >
> > Thanks for the comments and clarifications.

---

### Official Review · Reviewer_RjhJ · 2022-07-11

**Rating:** 8
**Confidence:** 3
**Soundness:** 4 excellent
**Presentation:** 4 excellent
**Contribution:** 3 good

**Summary:**

In the setting of active PAC learning, the algorithm has access to (cheap) unlabeled data and can make (more expensive) queries for their labels adaptively. For the hypothesis of polynomial threshold functions (PTFs) considered in this work, the queries are more complex, asking for the derivatives of the underlying polynomial. In order to measure the adaptivity, the algorithm may send the queries in rounds. The paper considers the variant of active learning known as perfect learning, where the goal for the learning algorithm is to label a set of $n$ (adversarially selected) samples. It also considers the setting where the data distribution and the underlying hypothesis come from known distributions rather than being worst-case.
The first result states that the query complexity as a function of $n$ is between $d \log n$ and $d^3 \log n$, where $d$ is the degree of the PTF. The second result gives an algorithm that makes the queries in batches, in the cost of incurring worse the query complexity. That algorithm has near-optimal runtime. The third result shows that missing access to any of the derivatives renders active learning of PTFs impossible. The next two results concearn learning in the average-case model: Upper and lower bounds on query complexity (where queries now only use labels) for the case where the samples and roots of the PTF are uniform in $[0,1]$ and upper bounds for PTFs with Dirichlet roots. Finally, the paper shows that when going to more than one variable, derivative queries are no longer sufficient to learn PTFs (in the worst-case setting).


**Questions:**

It would be nice to expand a bit more on the infinite precision problems of prior work (lines 181-183) and in what sense derivative queries are better.

**Limitations:**

The limitations are sufficiently addressed.

**Strengths And Weaknesses:**

Significance: There is a substantial body of work studying active learning of halfspaces, one of the most basic hypothesis classes considered in learning theory. Moving to polynomial threshold functions, a generalization of halfpsaces is a natural and well-motivated direction. The upper and lower bounds in the paper the paper give a fairly complete answer to an interesting question.

Quality: This review is mostly based on the short version of the paper and parts of the supplementary material corresponding to Sections 3 and 4. The key lemma for the wort-case active learning is Lemma 3.1 which finds partition in which every interval has the same sign pattern. The algorithm and coupon-collecting-based analysis for the average-case results seem fairly natural. The part about learning polynomials with Dirichlet roots needs considerable technical effort. Overall, the proofs I checked seem correct, although I have not gone through all the details.

Clarity: The paper is well-written. Below line 251, ‘d’ is undefined (though it is defined in the supplementary material) .

Originality: Not all of the parts of the paper appear to be equally challenging, but overall the arguments are non-trivial and require novelty. I am not an expert in the topic/prior work but the contributions seem solid.

Overall, a good theoretical work. I recommend that the paper be accepted.

---

> ### Author Response · Authors · 2022-07-29
> **Rebuttal**
>
> We thank the reviewer for their thorough review, and will add more discussion of issues with prior query types from the literature that require infinite precision in the final version.
>
> To clarify the comment from lines 181-183, the simplest known query type for learning the embedded linear version of the problem is a `generalized comparison,' which can ask for an arbitrary real number $\alpha \in \mathbb{R}$ and points $x_1$,$x_2$ whether $p(x_1) \geq \alpha p(x_2)$. This leads not only to issues in precision, but necessitates working with an infinite query set and in an algebraic computation model. The generic inference dimension based framework also breaks down in this setting, and the only known learning algorithms are substantially more complicated (e.g., relying on deep results in high-dimensional geometry by Barthe [Bar98]). Since the inference dimension framework is no longer used, this also means desirable traits that we got "for free" through the framework, such as computational efficiency and ease of analysis in the batch setting, are no longer guaranteed to hold.
>
> Finally, infinite query sets such as generalized comparisons also raise a substantial barrier to any hope of practical application: learning a PTF with generalized comparisons might require asking a doctor whether Patient A is $\pi$ times as sick as Patient B. One cannot expect such precision even in automated applications such as experimental design---whereas in comparison, asking for the sign of a derivative, especially a first derivative (does the patient become sicker with time?) or second derivative (does patient recovery accelerate as time passes?), seems more reasonable.

---

> > ### Comment · Reviewer_RjhJ · 2022-08-09
> > **Response acknowledgment**
> >
> > Thank you for the detailed response.

---

### Official Review · Reviewer_LNSA · 2022-07-16

**Rating:** 6
**Confidence:** 4
**Soundness:** 4 excellent
**Presentation:** 3 good
**Contribution:** 3 good

**Summary:**

The authors in this work study the problem of learning univariate PTFs. They consider the active learning model. Active learning is generally, the model that asks specific labels of points. The authors, show that in fact if we just ask for the labels we cannot get any useful guarantees. So, the authors are considering a stronger model, which asks for the sign of the derivative in a specific point and they show that again any learner that has some advantage has to query until the d derivative (d is the degree of ptf).

Their main result is an upper and a lower bound in the number of required queries in this model. The lower and the upper bounds are differ in a $d^2$ factor.

They also showed that if the roots of a PTF is uniformly distributed in [0,1] then again the same lower bound holds, but they can obtain a better upper bound that only losses $\tilde(O)(d)$ multiplicative factor.

Finally, they show that if we increase the dimension, then derivative queries cannot learn PTFs.



**Questions:**

I would like to know the authors opinion about my question above.

**Limitations:**

no limitations

**Strengths And Weaknesses:**

1. Overall, I find the results very interesting, especially the lower bounds.
2. Even if the upper bounds are not matching the lower bounds, the upper bounds are using several nice and simple ideas and they interesting on their own.
3. The authors showed that the fact that they have no distributional assumptions is the reason they required such a strong model. When the distribution and the ptfs are from a nice distribution, then they do not need such a strong model. It would be interesting to see when this threshold phenomenon occurs, what are the milder assumptions that we do not need derivatives.

Weaknesses/Questions:
1. The authors are working on a very strong model: Perfect Learning. But what happens in the Active Learning model with distributional assumptions only? Do label queries suffice? Do we need at least one derivative to get $\log(1/\epsilon)$?

Other:
I suggest the authors to put a definition of perfect learning in an environment (lines 98-101).

---

> ### Author Response · Authors · 2022-07-29
> **Rebuttal**
>
> We thank the reviewer for their question. Below we address the question more extensively, but in brief, the answer is positive: **derivatives are necessary in the standard active setting in (roughly) the same situations as in perfect learning even when the marginal data distribution is known.**
>
> In particular, active learning PTFs without derivatives requires $\Omega(1/\varepsilon)$ queries even when the data is drawn from some known distribution (e.g. uniform over $[0,1]$).
>
> To respond in more depth, we first note that perfect and active learning are classically related in many settings, where dependencies on $n$ in the query complexity for perfect learning typically translate into a $1/\varepsilon$ term for active learning. Below we dive into further detail; notably, the reviewer's question can be interpreted in multiple different ways depending on the nature of the distributional assumptions taken. The three possible interpretations are as follows, and we shall dedicated a paragraph to each of them:
> 1. "Hardest" distributional regime: The distribution-free setting for active learning (closest in spirit to worst case perfect learning). This setting refers to the case where the data is generated from an *arbitrary* distribution, entirely unknown to the learner (see, e.g., Theorems 1.1-1.3 in the paper).
> 2. "Intermediate" distributional regime, where we make assumptions only on the marginal distribution over data points.
> 3. "Easiest" distributional regime, where we make assumptions on both the marginal distribution and the hypotheses (see, e.g., Theorems 1.4-1.5 in the paper).
>
> **Hard setting:** In the distribution-free setting, it is well known that perfect and active learning are equivalent. This was shown in [KLMZ17]. Thus all our upper and lower bounds on `worst-case' perfect learning transfer directly to distribution-free active learning as well (where, as mentioned, all appearances of $n$ in our bounds should be replaced with $1/\varepsilon$, for example the $\log n$ term in Theorem 1.1 should be $\log 1/\varepsilon$ for active learning).
>
> **Intermediate setting:** In the intermediate setting where an assumption is made on the marginal distribution only, we do not know of a general equivalence between perfect and active learning (though the latter is still always `easier' than the former). In the case of PTFs, however, it is actually possible to give a reduction in the reverse direction as well, meaning active learning PTFs is essentially as hard as perfect learning, and requires $\Omega(1/\varepsilon)$ queries (while, in view of the ``$n$ versus $1/\varepsilon$ '' relation above, perfect learning requires $\Omega(n)$ queries). To see this, consider the setting in which the marginal data is drawn uniformly from $[0,1]$, but the adversary may choose an arbitrary quadratic (the argument easily extends to other reasonable distributions). As such, the adversary can choose from the set of quadratics with roots at $\{i\varepsilon, (i+1)\varepsilon\}$ for any $i \in [1/\varepsilon]$. To learn up to $\varepsilon/2$ error, it is necessary to distinguish between these hypotheses, or equivalently, to learn the sign of one point from each interval $I_i=[i\varepsilon,(i+1)\varepsilon]$. This is equivalent to hardness of perfect learning $1/\varepsilon$ points (albeit with a spreadness condition). It is not hard to show a $\Omega(1/\varepsilon)$ lower bound for the latter perfect learning problem (analogous to Dasgupta's classical halfspace lower bound [Das05]), and therefore also for the standard active setting if only label queries are allowed. Note that this also means the *intermediate setting for quadratics is essentially as hard as worst-case in most reasonable settings*.
>
> **Easy setting**
> Like the intermediate setting, there is no known reduction in the `easy' case when distributional assumptions are made jointly on the hypothesis and marginal. However, it is possible to show matching bounds in the particular settings we study. Upper bounds transfer directly from the perfect model (roughly replacing $n$ with $1/\varepsilon$), and the lower bounds transfer similarly due to the fact that the same entropic analysis can be used to bound query complexity in the active case (after suitably discretizing the problem). As such, the models are again equivalent in the cases we study, and analogues of Theorems 1.4 and 1.5 hold for active learning with a $1/\varepsilon$ term replacing essentially each $n$ term. However, indeed as the reviewer points out, this may not always be the case under (other) weaker assumptions on the joint distribution, and studying threshold phenomena in this setting is certainly an interesting problem.

---

> > ### Comment · Area_Chair_fyc1 · 2022-08-08
> > **Please acknowledge the authors' rebuttal**
> >
> > Please acknowledge the authors' rebuttal.

---

### Author Response · Authors · 2022-07-29
**Rebuttal (Joint Response)**

We thank all reviewers for their thorough reading of our work and helpful comments and questions. Below, we address two commonly asked questions (tightness of worst-case bounds and the multivariate setting), which we believe are arguably the two most interesting future research directions suggested by our work. We also respond to each reviewer individually in separate comments under each review.

**Multivariate case:** Theorem 1.6 in our work shows that derivative queries do not suffice, in the worst case, for efficient perfect (or active) learning even for bivariate PTFs. A natural question asked by multiple reviewers is whether we anticipate that positive worst case results (similar to those from the univariate case) hold in the multivariate case under a stronger query model. Our hunch is that the answer is positive; while we do not have any concrete results at this point, we conjecture that queries that involve computing signs of integrals of the gradient over suitable $(k-1)$-dimensional manifolds in the $k$-variate case (e.g., taking the integral of the gradient over a curve, for example a straight line, in the plane) might be strong enough to allow efficient perfect/active learning PTFs.

It is also worth observing that our multivariate lower bound relies on a fairly adversarial configuration of points. It is entirely possible that under strong enough distributional assumptions, standard label queries or label queries with low-order derivatives suffice (similarly to our results in the univariate case). Better understanding this regime remains an intriguing open problem.

**Tightness:** The second main problem left open by our work concerns the tightness of our worst-case bounds on inference dimension (between $\Omega(d)$ and $O(d^2)$) and perfect/active learning itself ($\Omega(d\log(n))$ vs. $O(d^3\log(n))$ for perfect learning). While we feel both questions could go either way, we believe it is unlikely that $O(d^3\log(n))$ is the correct answer. Historically it has often been the case that related problems, after many years of effort, do manage to achieve the information-theoretic lower bound. This was the case for halfspaces in many settings, where achieving the analogous $O(d\log(n))$ bound in the worst-case (called the `point-location' problem) took nearly 40 years. It is possible that in this case the correct bound is $\Theta(d^2\log(n))$ due to the added need for derivatives, but $\Theta(d^3\log(n))$ seems unlikely to be the final answer.

---

### Meta-Review · Area_Chair_fyc1 · 2022-08-23

**Recommendation:** Accept
**Confidence:** Certain

**Metareview:**


This paper contains a fresh and mathematically interesting theoretical analysis of a fundamental problem.

**Award:**

Yes

---

### Decision · Program_Chairs · 2022-09-14

Accept